# Self-Similarity of Continuous-Spectrum Radiative Transfer in Plasmas with Highly Reflecting Walls

**Alexander B. Kukushkin [1,2,3,*] and Pavel V. Minashin [1]**

[1] National Research Center "Kurchatov Institute", 123182 Moscow, Russia; Minashin_PV@nrcki.ru
[2] National Research Nuclear University MEPhI (Moscow Engineering Physics Institute),
115409 Moscow, Russia
[3] Moscow Institute of Physics and Technology, National Research University, 141700 Dolgoprudny, Russia
[*] Correspondence: Kukushkin_AB@nrcki.ru

**Abstract:** Radiative Transfer (RT) in a continuous spectrum in plasmas is caused by the emission and absorption of electromagnetic waves (EM) by free electrons. For a wide class of problems, the deviation of the velocity distribution function (VDF) of free electrons from the thermodynamic equilibrium, the Maxwellian VDF, can be neglected. In this case, RT in the geometric optics approximation is reduced to a single transport equation for the intensity of EM waves with source and sink functions dependent on the macroscopic parameters of the plasma (temperature and density of electrons). Integration of this equation for RT of radio-frequency EM waves in laboratory plasmas with highly reflecting metallic walls is substantially complicated by the multiple reflections which make the waves with the long free path the dominant contributors to the power balance profile. This in turn makes the RT substantially nonlocal with the spatial–spectral profile of the power balance determined by the spatial integrals of the plasma parameters. The geometric symmetry of the bounding walls, especially when enhanced by the diffuse reflectivity, provides a semi-analytic description of the RT problem. Analysis of the accuracy of such an approach reveals an approximate self-similarity of the power balance profile and the radiation intensity spectrum in both approximate and ab initio modeling. This phenomenon is shown here for a wide range of plasma parameters and wall reflectivity, including data from various numeric codes. The relationship between the revealed self-similarity and the accuracy of numeric codes is discussed.

**Keywords:** radiative transfer; continuous spectra; electron-cyclotron radiation; thermonuclear fusion plasma; tokamak-reactor; ITER

## 1. Introduction

Self-similarity phenomena in the theory of the radiative transfer in plasmas play an important role in identifying the main scalings and elaborating on the approaches to solving the problems of time-consuming numerical modeling. An example of such self-similarity is presented in [1], where it is shown that the Green's function of the non-stationary radiative transfer (RT) in the spectral lines of atoms and ions in plasma and gases has an approximate self-similarity in a wide range of RT problems. In the case of electromagnetic radiation in spectral lines, the radiation is emitted and absorbed by bound electrons in atoms and ions. Another type of RT is associated with radiation emission and absorption by free electrons in plasmas. In this case, RT takes place in the continuous spectrum of electromagnetic waves. The difference in the kinetics of free and bound electrons leads to a very different description of RT.

Here we consider the self-similarity of the power balance profile and the intensity of escaping radiation for continuous-spectrum radiative transfer in plasmas with highly reflecting walls. The motivation of this research is based on the need for massive predictive modeling of the electron cyclotron radiation (ECR) transport in experimental facilities for magnetic confinement of hot plasmas in a wide range of plasma temperatures, including

temperatures in a future thermonuclear fusion reactor such as the ITER tokamak [2] (under construction) and various next-step projects called DEMO (see, e.g., [3]).

The crucial importance of the ECR power loss in magnetic thermonuclear fusion was realized at an early stage of research. The first estimates of the power loss in two alternative limits, namely volumetric losses (which corresponds to neglecting the absorption of emitted radiation) and surface losses (thermodynamic-equilibrium, black-body intensity of the outgoing radiation due to imprisonment of radiation at high values of the optical thickness of the plasma), showed the impossibility of fulfilling the criteria for thermonuclear ignition in a hot homogeneous plasma in a laboratory. However, the actual ECR power loss appeared to be much less than the above limits. An accurate analysis of the problem required the development of a complex theory of ECR transport [4–9] (due to the strong angular and frequency dependence of the emission and absorption functions on the plasma temperature [4,9]). In particular, it was shown that for the expected parameters of a magnetic thermonuclear fusion reactor (e.g., torus major radius $R_0$ = 6 m, minor radius $a$ = 2 m, elongation $k_{\text{elong}}$ = 1.0, magnetic field on torus axis $B_0$ = 5 T, volume-average electron temperature $<T_e>_V$ = 20 keV and density $<n_e>_V$ = $10^{20}$ m$^{-3}$, see Figure 1), only a small fraction (~0.01 $\sqrt{1-R_w}$, $R_w$ is the wall reflection coefficient) of the emitted ECR power escapes from the plasma volume [5,8] (see, e.g., Figure 6 in [5] and Figures 4 and 5 in [8] for the "transparency factor" that quantifies the aforementioned fraction). In present-day magnetic confinement facilities, the internal ECR (i.e., not an external one, which is injected into the plasma for its auxiliary EC resonance heating (ECRH) and/or EC current drive (ECCD)) plays an important role only for diagnostics of the electron temperature, and not for spatially local and total power balance.

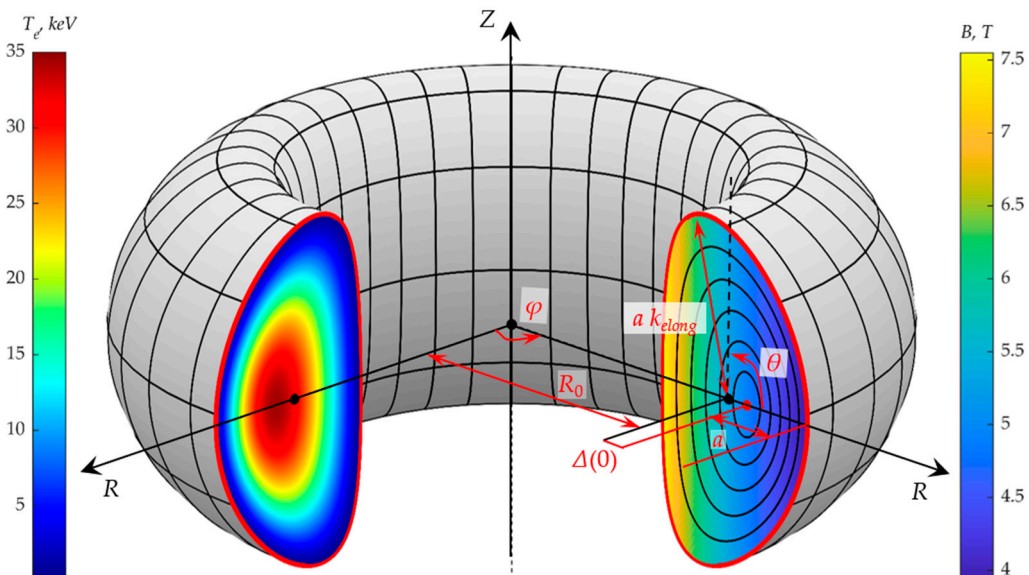

**Figure 1.** Geometrical parameters, magnetic surfaces structure, and total magnetic field profile (on the right poloidal cross-section of the toroid) and respective profile of electron temperature (on the left poloidal cross-section of the toroid). $R_0$, $a$—major and minor radii of the plasma column, $k_{\text{elong}}$—elongation, $\theta$, $\varphi$—poloidal and azimuthal angles, and $\Delta(0)$—the shift of magnetic axis with respect to the vessel's toroidal axis (Shafranov shift). Magnetic surfaces are calculated with (3) for the following given moments: magnetic surface radius, $a_{metr}(\rho)$, Shafranov shift, $\Delta(\rho)$, triangularity, $\delta(\rho)$, and vertical elongation, $\lambda(\rho)$, which are taken from the ASTRA code calculations of quasi-steady-state ITER-like scenario.

For the next generation of tokamaks like ITER and DEMO, because of expected high temperatures in the central plasma and strong magnetic field, the ECR power loss will play an important role in the balance of electron energy (see [10–12] for ITER and [3] for DEMO), and will also be a source of thermal and electromagnetic loads on in-chamber components and diagnostic tools [13]. Modelling of quasi-steady state regimes of operation

predicts a significant contribution of ECR power loss to the local balance of electron power in ITER [11,12,14] and DEMO [3]. The ECR power loss can also limit the temperature excursions of the thermonuclear fusion power in ITER and DEMO for a central electron temperature $T_e(0) > 30$ keV [15]. The foregoing required the development of numerical codes for more accurate calculations of the ECR transport (especially in the central plasma).

The geometric symmetry of the bounding walls, namely the toroidal symmetry of the vacuum chamber, with a sufficiently high reflectivity, made it possible to create a simple fast-routine method [16], which provides the calculation of the spatial profile of the ECR power balance with a sufficiently high accuracy. The idea of this approach, implemented in the CYTRAN code [16], is based on the analysis of the results of three-dimensional (3D) Monte Carlo modeling of ECR transport using the SNECTR code [17]. Further modification of the method [16] in [18–21], the benchmarking [22] of all existing codes for ECR transport in tokamak-reactors (including SNECTR, CYTRAN, CYNEQ [10,18], and EXACTEC [23] codes) and additional comparison of codes in [12,24] (including comparison with the latest code, RAYTEC [25]) clarified the status of fast-routine approaches for use in mass predictive modeling of tokamak-reactor operation. The problem of self-similarity of the spatial profile of the ECR power balance, which we analyze, is the next step in benchmarking of codes.

## 2. Materials and Methods

### 2.1. Basic Equations

Electromagnetic (EM) waves in plasmas are described by a system of self-consistent equations, which includes Maxwell's equations for the classical (non-quantized) electromagnetic field in plasmas and kinetic equations (i.e., statistical equations of motion) for various types of plasma particles.

From Maxwell's equations, one can obtain the radiation transfer equation in the approximation of the ray trajectory description of wave propagation (the so-called geometric optics approximation) [26–28]:

$$\left( \frac{1}{v_g} \frac{\partial}{\partial t} + \mathbf{n} \frac{\partial}{\partial \mathbf{r}} \right) \frac{J(\phi, \mathbf{r}, t)}{N_r{}^2} = \frac{q(\phi, \mathbf{r}, t)}{N_r{}^2} - \kappa(\phi, \mathbf{r}, t) \frac{J(\phi, \mathbf{r}, t)}{N_r{}^2}, \tag{1}$$

where $\phi = \{\omega, \mathbf{n}, \xi\}$ are the parameters of the EM wave: $\omega$—frequency, $\mathbf{n} = \mathbf{k}/k$—unit vector of propagation direction, $\mathbf{k}$—wave vector, $\xi$—the type of the wave (i.e., polarization of the wave); $J(\phi, \mathbf{r}, t)$—radiation intensity (i.e., energy flux density, differential with respect to all parameters of the EM wave); $q(\phi, \mathbf{r}, t)$—the power density of the spontaneously emitted EM waves (usually called the source function or emissivity); $\kappa(\phi, \mathbf{r}, t)$ is the coefficient of absorption of the wave by the medium (the reciprocal of the free path for the given wave parameters), in the calculation of which the stimulated emission is also taken into account; $N_r$—ray refractive index, and $v_g$—group velocity of EM waves. The absorption coefficient, $\kappa$, and emissivity, $q$, contain averaging over the electron velocity distribution function; therefore, in cases where radiation can distort the electron distribution function, the functions $\kappa$ and $q$ implicitly depend on the radiation intensity, $J$.

Here, the stationary case will be considered for the ECR transport equation, since the characteristic time for the establishment of the stationary intensity is much shorter than the time during which the macroscopic parameters of the medium (temperature, electron density, and external magnetic field) change.

If the influence of radiation on the velocity distribution function of emitting electrons can be neglected (i.e., assuming that the velocity distribution function is "frozen"), Equation (1) becomes closed. It can be solved by integrating along the path of the rays of the EC waves in the plasma, taking into account the absorption of ECR by the medium:

$$J(\phi, s) = J_0(\phi, s) \exp\left(-\int_{s_0}^{s} \kappa(\phi, s')ds'\right) + \int_{s_0}^{s} \frac{q(\phi, s')}{N_r{}^2} \left[ \exp\left(-\int_{s'}^{s} \kappa(\phi, s'')ds''\right) \right] ds', \tag{2}$$

where the integrals are taken along the ray connecting the points with coordinates $s_0$ and s along this ray.

In a tokamak, which is a toroidal axisymmetric system with nested magnetic flux surfaces, one has to solve (2) with account of multiple reflections of the EC radiation from the metallic wall of the vacuum chamber. The geometry of a plasma in the state of force equilibrium may be described in the 3-moment approximation [29]:

$$
\begin{cases}
R(\rho,\theta) = R_0 + \Delta(\rho) + a_{\text{metr}}(\rho)\,(\cos\theta - \delta(\rho)\sin^2\theta), \\
\qquad\quad Z(\rho,\theta) = a_{\text{metr}}(\rho)\,\lambda(\rho)\,\sin(\theta)\,.
\end{cases}
\tag{3}
$$

where $R_0$ is the major radius of torus, $a_{metr}$ is the half width of the magnetic surface in the equatorial plane of the torus, $\Delta(\rho)$ is the Shafranov shift, $\lambda(\rho)$, and $\delta(\rho)$ are the vertical elongation and triangularity of the magnetic surface (see Figure 1). These momenta and the two-dimensional magnetic field are taken from the plasma equilibrium, calculated self-consistently in the 1.5D transport simulations (2D plasma equilibrium and 1D transport) carried out using the ASTRA suite of codes [30,31] for solving the system of equations for the force equilibrium of the plasma and the heat and particle transport. An example of the topology of magnetic flux surfaces is shown in Figure 1 for the case of a quasi-steady-state ITER-like scenario which is close to the ITER operation scenario considered in [32].

We will analyze the local power loss density, $P_{\text{EC}}(\rho)$, calculated from the balance of absorption and emission of ECR, averaged over magnetic surface:

$$
P_{\text{EC}}(\rho) = \left\langle \sum_{\xi} \int d\omega \int d\Omega_{\mathbf{n}}\ \left[q_{\xi}(\rho,\,\theta,\omega,\mathbf{n}) - \kappa_{\xi}(\rho,\theta,\,\omega,\mathbf{n})\,J(\omega,\mathbf{n},\xi,\rho,\theta)\right] \right\rangle_{\text{ms}},
\tag{4}
$$

For the plasma in local thermodynamic equilibrium (Maxwellian velocity distribution of electrons in plasma), according to Kirchhoff's law for the relationship between the absorption coefficient and emissivity, Equation (4) becomes:

$$
P_{\text{EC}}(\rho) = \left\langle \sum_{\xi} \int d\omega \int d\Omega_{\mathbf{n}}\ \kappa_{\xi}(\rho,\theta,\,\omega,\mathbf{n})\ \left[J_{BB}(\omega,\mathbf{n},\xi,\rho) - J(\omega,\mathbf{n},\xi,\rho,\theta)\right] \right\rangle_{\text{ms}},
\tag{5}
$$

$$
J_{BB}(\omega,\mathbf{n},\xi,\rho) = \frac{\omega^2\,T_e(\rho)}{8\pi^3 c^2},
\tag{6}
$$

where $J_{BB}$ is blackbody intensity and $c$ is the speed of light in vacuum.

The averaging over magnetic surface is expressed as follows:

$$
F(\rho) \equiv\, <F(\rho,\theta)>_{\text{ms}} = \left[\int_0^{2\pi} F(\rho,\theta)\,g(\rho,\theta)\mathrm{d}\theta\right] \cdot \left[\int_0^{2\pi} g(\rho,\theta)\mathrm{d}\theta\right]^{-1}
\tag{7}
$$

where $g(\rho,\theta)$ is the plasma volume inside the surfaces $[\rho \div \rho + d\rho,\ \theta \div \theta + d\theta]$, so that the volume averaging is determined by the formula:

$$
\begin{aligned}
<F>_{\text{v}} &= \frac{\int_0^1 \int_0^{2\pi} F(\rho,\,\theta)\,g(\rho,\theta)d\rho d\theta}{V_{\text{tot}}}, \\
V_{\text{tot}} &= \int_0^1 \int_0^{2\pi} g(\rho,\theta)d\rho d\theta,
\end{aligned}
\tag{8}
$$

where $V_{\text{tot}}$ is the total volume of toroidal plasma.

Total (i.e., volume-integrated) EC power loss is calculated as follows:

$$
P_{EC}^{tot} = \int_0^1 P_{\text{EC}}(\rho)\,\frac{dV}{d\rho}\,d\rho
\tag{9}
$$

$$\frac{dV}{d\rho} = \int_0^{2\pi} g(\rho,\theta)d\theta \qquad (10)$$

The local magnetic field in the plasma, expressed in terms of toroidal and poloidal components, $B^2 = B_{tor}^2 + B_{pol}^2$, can be derived from the plasma equilibrium or taken constant:

- 2D approximation: magnetic field, $B(\rho,\theta)$, is a function of the normalized toroidal magnetic flux within magnetic surface, $\rho$, and of the poloidal angle, $\theta$:

$$B = B(\rho,\theta), \qquad (11)$$

- 1D approximation: one-dimensional profile, $B(\rho)$, is derived by averaging the field $B(\rho,\theta)$ over magnetic surface:

$$B = B(\rho), \qquad (12)$$

- 0D approximation: homogeneous profile, $B = $ const, may be taken as $B_{tor}(R_0) \equiv B_0$, where $B_0$ is the vacuum toroidal magnetic field at $R_0 = (R_{max} + R_{min})/2$, or as the volume-averaged total magnetic field, $<B>_v$:

$$B = B_0 = \text{const.} \qquad (13)$$

The profile of magnetic field distribution over magnetic flux surfaces, which is obtained from the 2D distribution $B(\rho,\theta)$ by averaging over each magnetic surface, appears to be rather flat, as it weakly depends on $\rho$. The latter justifies the use of the 0D approximation for the magnetic field in some cases (see below). For the 1D profiles of electron density and temperature in the case close to the ITER operation scenario [32], the 1D profile of the magnetic field is shown in Figure 2.

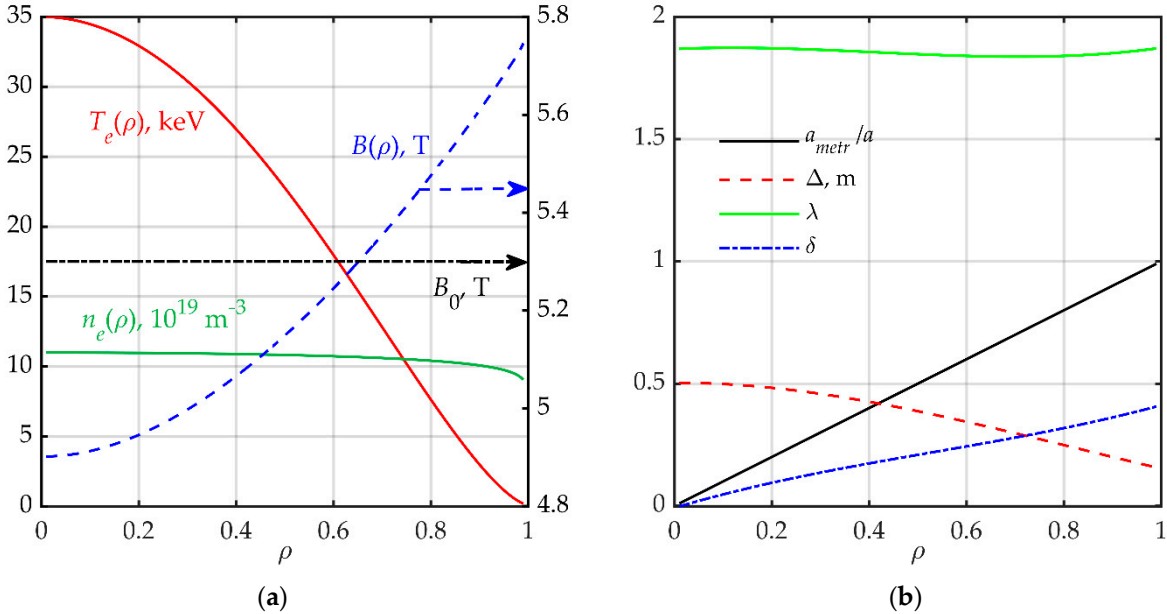

(**a**)  (**b**)

**Figure 2.** Parameters of plasma for the ITER-like scenario: (**a**) profiles of the electron temperature, $T_e(\rho)$, and density, $n_e(\rho)$, 1D (i.e., averaged over magnetic surface) total magnetic field, $B(\rho)$, and vacuum magnetic field on the axis, $B_0$; (**b**) profiles of the moments of magnetic surfaces: half-width of the magnetic surface, $a_{metr}$, Shafranov shift, $\Delta$, elongation, $\lambda$, and triangularity, $\delta$. Equilibrium is calculated using the ASTRA code.

Figure 1 shows that the spatial profile of magnetic field is strongly inhomogeneous so that the transport of the ECR, which is characterized by strong spectral and angular dependences of the absorption coefficient $\kappa$ and the source function $q$ (see, e.g., Equations (3.11),

(3.4), and (3.13) in [4] and (3.1.73)–(3.1.78) in [9]), can strongly depend on the 2D distribution of the magnetic field strength $B$, and only direct 3D modeling of ray trajectories with account of multiple reflections from the wall can adequately describe the problem. It turns out, however, that the symmetry of the highly reflecting wall of the vacuum chamber substantially influences the angular and spatial distribution of the radiation intensity.

*2.2. Various Approaches to Solution of ECR Transport Problem in Toroidal Plasmas with Highly Reflecting Walls and Their Comparison*

The first numerical simulation of the ECR transport in toroidal plasma, which is confined by a magnetic field in a vacuum chamber with highly reflecting walls, was carried out using the SNECTR code [17], which provides Monte Carlo simulations of EC wave ray trajectories under the following conditions:

- hot Maxwellian plasma with a volume-averaged electron temperature of $\langle T_e \rangle_V \geq 10$ keV;
- toroidal plasma with a noncircular cross section and moderate aspect ratio (i.e., ratio of tokamak major $R_0$ to minor radius, $a$, see Figure 1) A ~ 3;
- multiple reflections of radiation from the wall of vacuum chamber.

Under the conditions indicated above, the contribution of ECR at low-number harmonics of the EC fundamental frequency ($n = 1$ and $n = 2$) can be neglected, and the effect of plasma on the EC ray trajectories can also be neglected (i.e., $N_r = 1$ in (1)). Also, the calculations using the SNECTR code were carried out in the 0D approximation (13) for the magnetic field (cf. Figure 2).

This approach made it possible to extend the previous results [5,6,8,9] for ECR transport in hot homogeneous plasmas to the case of strongly inhomogeneous plasmas in a facility with highly reflective walls. It was found that under these conditions the spatial profile of the power balance, $P_{EC}(\rho)$, defined in (4), changes the sign in the region of the cold peripheral plasma: here, the plasma turns out to be an effective absorber of the radiation emitted in the hot central plasma, so that the net power balance at the periphery is negative. Analysis [16] of the results from SNECTR code has shown that in the range of radiation frequency, which is responsible for the main contribution to $P_{EC}(\rho)$, radiation intensity is almost isotropic in angles of the wave vector and almost homogeneous in $\rho$.

The assumption of angular isotropy of the radiation intensity and the use of the 0D approximation for the magnetic field greatly simplifies the ECR transport problem: under this assumption the profile of the EC power loss can be described by a 1D function that depends only on the coordinate of the magnetic flux surface in toroidal plasma.

In the CYTRAN code [16], the ECR transport depends only on the angle-averaged values of the absorption coefficient, $\kappa(\mathbf{r}, \Phi)$, and the emissivity, $q(\mathbf{r}, \Phi)$:

$$\kappa_{\xi}(\mathbf{r}, \omega) \equiv \int \frac{d\Omega_{\mathbf{n}}}{4\pi} \kappa(\mathbf{r}, \Phi), \quad q_{\tilde{\xi}}(\mathbf{r}, \omega) \equiv \int \frac{d\Omega_{\mathbf{n}}}{4\pi} q(\mathbf{r}, \Phi), \tag{14}$$

CYTRAN uses approximate formulae for functions (14), that have satisfactory accuracy only for high temperatures, $T_e \geq 10$ keV.

The next simplification of the RT problem in the CYTRAN code is related to the division of the plasma into an optically-thick inner region and an optically-thin outer region. The thick–thin interface is defined in terms of a critical optical thickness, $\tau$, along the radial coordinate. The choice of $\tau$ value of the order of unity was justified by the best agreement with the results of the SNECTR code simulations. In the semi-analytic approach of CYTRAN, the density of the ECR power loss, $P_{EC}(\rho)$, is determined by the following power balance. The power escaping from the inner region is equal to the black body losses from the surface of the inner region with the temperature at the interface between the inner and outer regions, divided by the total volume of the inner plasma. The power balance in the outer region is determined by the emissivity and absorption, both integrated over the volume of the outer region, and by the power outgoing into the inner region and exiting through the wall, with account of the reflection from the wall. This model has proven to be very successful in describing the results for $P_{EC}(\rho)$ from 3D modeling using

SNECTR. However, in CYTRAN, the $P_{EC}(0)$ value diverges, and it was necessary to correct the formalism in the central plasma.

The modification of the method [16] was made in [18–21] and implemented in the CYNEQ code. In [18,19], the model of CYTRAN code for the inner region (optically-thick plasma) was corrected with the following simplification: the use of the Escape Probability method developed for the RT in spectral lines of atoms and ions (see References [13,17,18] in [1]) to the case of RT by free electrons suggested an approximation, which gives $P_{EC}(\rho,\omega) = 0$ in the inner optically-thick region. This approach eliminated the divergence of $P_{EC}(\rho)$ at $\rho = 0$ and gave the results for $P_{EC}(\rho)$ which are in good agreement with the results of the SNECTR code. In addition, absorption coefficient (14) in the CYNEQ code is calculated by numerical averaging the Schott–Trubnikov emissivity of a single electron [4,26] over an arbitrary electron velocity distribution. Comparison of different approaches to calculating (14) is carried out, for example, in the benchmarking [22] of the codes (see Figure 4 and Equation (5) in [22]).

Further progress was made in [20]: the description of magnetic field was extended to the case of 1D and 2D approximations. In the 2D geometry, absorption and emission coefficients can be expressed as functions of normalized radius, $\rho$, poloidal angle, $\theta$, and normalized frequency, $\widetilde{\omega} = \omega/\omega_{B0}$ ($\omega_{B0} = eB_0/m_e c$ is the fundamental EC frequency, where $B_0$ is the vacuum toroidal magnetic field on tokamak's toroidal axis, $B_0 = B(R = R_0, Z = 0)$, cf. Figure 1):

$$\kappa_\xi(\rho,\ \theta,\ \widetilde{\omega}) = \frac{\omega_{pe}^2\ B_0}{c\ \omega_{B0}\ B(\rho,\ \theta)} \chi_\xi\left(\frac{\widetilde{\omega}\ B_0}{B(\rho,\ \theta)},\ T_e(\rho)\right) \tag{15}$$

$$q_\xi(\rho,\ \theta,\ \widetilde{\omega}) = \kappa_\xi(\rho,\ \theta,\ \widetilde{\omega})\ \frac{\widetilde{\omega}^2\ \omega_{B0}^2\ T_e(\rho)}{8\ \pi^3 c^2}. \tag{16}$$

where $\omega_{pe}$ is the plasma frequency and function $\chi_\xi$ depends only on the local value of normalized frequency and the local value of the electron temperature. Similar to CYTRAN code, the phase space $\Gamma = \{\mathbf{r},\ \omega,\ \xi\}$, where $\mathbf{r}$ is the spatial coordinate and $(\omega,\ \xi)$ are the wave's parameters, is divided into two parts according to the type of ECR transport: (i) an optically-thin layer of outer plasma (where the transport is nonlocal):

$$\Gamma_{esc} = \left\{(\rho,\ \theta,\ \omega,\ \xi):\ \int_\rho^1\ d\rho\ a\cdot\kappa_\xi(\rho,\ \theta,\ \omega) \le \tau_{crit} \cong 1\right\} \tag{17}$$

and (ii) an optically-thick inner part of plasma with dominant diffusive transport. (In general, in a wide range of frequencies, essential for $P_{EC}(\rho)$ formation, the optically-thin zone can cover the entire plasma volume, see Figure 3). Formula (17) defines the boundary between these parts, where the function $\rho_{cut}(\omega,\ \theta,\xi)$, in contrast with [10,18], depends on the poloidal angle (cf. [33]). The intensity of the EC radiation, escaping from the plasma, is determined by the nonlocal part of the phase space:

$$J_{esc}(\omega,\xi) = \frac{\langle q_\xi(\rho,\ \theta,\ \omega)\rangle_{V_{esc}}}{\int \frac{d\Omega_\mathbf{n}}{4\pi} \int \left(\mathbf{n}, \frac{d\mathbf{S_w}}{V_{esc}}\right)(1 - R_w) + \langle\kappa_\xi(\rho,\ \theta,\omega)\rangle_{V_{esc}}}, \tag{18}$$

where $V_{esc}$ is a projection of phase space of Equation (17) to its coordinate part; $\langle\ \ \rangle_{V_{esc}}$ denotes averaging over the volume, defined in (8); $S_w$ is the inner, plasma-facing surface of the vacuum vessel; $R_w$ is the coefficient of reflection of waves from this surface, which generally is a function of the frequency, $\omega$, and the direction of the wave, $\mathbf{n}$. Thus, in Equation (4), intensity can be expressed as follows:

$$J(\omega,\xi,\rho,\theta) = J_{esc}(\omega,\xi)\eta(\rho - \rho_{cut}(\omega,\ \theta,\xi)) + J_{BB}(\omega,T_e(\rho))\eta(\rho_{cut}(\omega,\ \theta,\xi) - \rho) \tag{19}$$

where $\eta$ is the Heaviside function. To improve the accuracy, the boundary between the intensity (18) and the blackbody intensity can be made smooth using an interpolation (see Equation (10) in [12]). An example of the spectral-spatial distribution of the ECR power loss for the conditions of ITER operation close to [32] is shown in Figure 3.

In the version of the CYTRAN code [21], published in National Transport Code Collaboration (NTCC) project, the algorithm [16] of calculating the power balance in the inner region (optically-thick plasma) is changed by means of normalization which substantially, by an order of magnitude and more, reduces the contribution of the inner region to $P_{EC}(\rho)$. This normalization is consistent with the model [18,19] where, according to the second term in (5) in the case of thermodynamic equilibrium, we have $P_{EC}(\rho,\omega) = 0$ in the region of optically-thick plasma.

The first benchmarking of codes for calculating the $P_{EC}(\rho)$ profile was carried out in [22] with the following codes: CYTRAN [16] with the aforementioned modification [21], CYNEQ [10,18], and EXACTEC [23]. Benchmarking was carried out in a wide range of temperature, $T_e(\rho)$, and density, $n_e(\rho)$, profiles expected in reactor-grade tokamaks, and for a flat profile of magnetic surface-averaged magnetic field, $B(\rho) = const$ (0D approximation of the magnetic field profile). The results were benchmarked versus predictions of (at that time) the most comprehensive code SNECTR [17,34] (currently not used), based on the Monte Carlo simulations of the EC emission and absorption processes in axisymmetric toroidal plasmas. A comparison of results was made for the cases which differ in the geometry of the plasma and the type of the reflection of the EC waves from the wall of the vacuum vessel. These cases were divided into the following two groups:

(A)   A cylinder with circular cross-section, specular reflection (this case was considered specifically for the EXACTEC code, which is applicable only for this geometry, and SNECTR code, which only applied to plasma with a circular cross-section)

(B)   (i) A circular cylinder with diffuse reflection (this case was covered only by the SNECTR code calculations); (ii) Any (cylindrical or toroidal) geometry with diffuse reflection or a noncircular toroid with any (specular or diffuse) reflection (all these subcases may only be covered by the CYTRAN and CYNEQ codes).

The above division of cases is determined by the following properties of radiative transfer under conditions of noncircular toroidal geometry or diffuse reflection:

- approximate spatial homogeneity of the radiation intensity, since the wave trajectories uniformly fill the plasma volume,
- approximate angular homogeneity of the radiation intensity, it is more isotropic in the angles of the wave vector in comparison with the case of specular reflection in a circular cylinder.

The approximate homogenization and isotropization of the radiation intensity for large enough $R_w$ occurs in a limited range of radiation frequencies (namely, in the range where the wave free path is comparable with, or exceeds, the radius of the plasma column, e.g., minor radius of toroidal plasma), however it is these frequencies that are responsible for dominant contribution to $P_{EC}(\rho)$ (see [10,16–18,23]). In case B, which is much closer to experimental conditions, the radiation emitted in the hot plasma core travels longer in a cold periphery, as compared with case A, and therefore it is absorbed there stronger so that the cold periphery appears to be the net absorber of radiation. This picture is very different from case A, where the radiation emitted in the hot plasma core is reflected from the wall back making the absorption in the core higher than in case B.

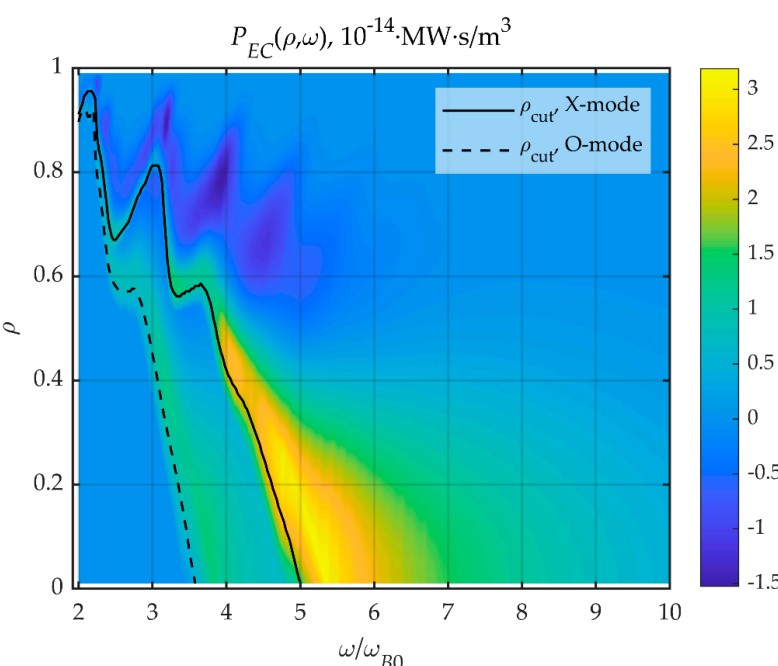

**Figure 3.** Spectral–spatial distribution of the EC power loss for ITER-like scenario with parameters shown in Figure 2 and wall reflection coefficient $R_w = 0.9$. Boundaries between optically-thick inner plasma and optically-thin outer plasma are shown for two polarizations of the EC waves in plasma: X-mode (solid line) and O-mode (dashed line).

The benchmarking [22] has shown good agreement of results within tasks A and B.

The further progress in simulations of EC power losses has been achieved in the RAYTEC code [25] via direct integrating along ray paths of EC waves in a toroid, instead of using the analytic solution [8] of radiative transfer problem for a circular cylinder with mirror reflections from the wall, which is used in EXACTEC. The RAYTEC code uses a 2D profile of magnetic field but does not take into account the effects of plasma equilibrium (Shafranov shift). The latter effects may be simulated using the modified CYNEQ code [12,14,20] which uses the 2D geometry of magnetic field $B(R, Z)$, the CYNEQ-B(2D) code. This code can work—with almost the same accuracy as the fast-routine code CYNEQ-B(1D)—as a part of the ASTRA code [31] that can perform self-consistent 1.5D transport simulations (2D force equilibrium and 1D transport).

Benchmarking of ECR transport codes carried out in [22] for the flat profile of the magnetic field averaged over magnetic flux surface, $B(\rho) = \mathrm{const}$, was appended in [12,24] with the results for self-consistent 2D plasma equilibrium and a comparison with the latest code RAYTEC [25]. The conclusion of these benchmarkings was that the modified codes CYNEQ [20] and CYTRAN [21] (in its version participated in the benchmarking [22], see below Section 3.1) are suitable for use in global transport codes (e.g., ASTRA [31]) for self-consistent 1.5D transport simulations of plasma evolution in tokamak-reactors because they provide good approximation and computational efficiency. A comparison of CYNEQ and CYTRAN results with the latest results [35] from RAYTEC simulations of ECR power loss density for DEMO-like high-temperature plasmas in the present paper confirms this conclusion.

### 2.3. Similarity of Spatial Distributions of Net EC Power Loss and Spectral Distributions of Radiation Intensity

The effect of the universal shape (self-similarity) of the normalized $P_{EC}(\rho)$ profiles for the same shapes of the electron temperature and density profiles and their significantly

different volume-averaged values, found in [36] and based on the results of CYNEQ-B(0D) calculations, is formulated as follows:

$$P_{EC}^{norm}(\rho) \equiv \frac{P_{EC}(\rho)}{P_{EC}^{tot}/V_{tot}} = f(\rho, \{T_e^{norm}(\rho)\}, \{n_e^{norm}(\rho)\}, R_w), \tag{20}$$

where $P_{EC}^{tot}$ is the total (i.e., volume-integrated) EC radiation power loss (9); $V_{tot}$ is the plasma volume; the brackets { } stand for a functional dependence; $T_e^{norm}(\rho)$ and $n_e^{norm}(\rho)$ are the normalized profiles of temperature and density:

$$T_e^{norm}(\rho) = \frac{T_e(\rho)}{< T_e >_V}, \tag{21}$$

$$n_e^{norm}(\rho) = \frac{n_e(\rho)}{< n_e >_V}. \tag{22}$$

The scaling law (20) appears to be valid for the results of calculations of various codes for the EC power loss density [37].

A similar effect has now been found for the spectral distribution of the intensity of the radiation escaping from the plasma (i.e., for the intensity (18)) for a wide class of normalized temperature profiles. To obtain a universal spectral shape of the escaping ECR intensity, we normalize the frequency as follows:

$$\overline{\omega} = \frac{\omega - \omega_{min}}{\omega_{max}(\{T_e(\rho)\}, \{n_e(\rho)\}, R_w)}, \tag{23}$$

where $\omega_{min} = 2\,\omega_{B0}$, $\omega_{max}$ is determined by the equation for the total EC power loss, obtained from the intensity spectra: upper limit of integration of the spectral intensity is chosen so that the integral over the frequency range $[\omega_{min}, \omega_{max}]$ coincides up to 1% with the exact integral over frequencies:

$$\int_{\omega_{min}}^{\omega_{max}} \sum_\xi J_{esc}(\omega, \xi)\, d\omega = \frac{P_{EC}^{tot}}{\pi S_{wall}(1 - R_w)} \tag{24}$$

Using this definition of $\omega_{max}$, the formula for the universal shape of the EC intensity will be as follows:

$$J_{esc}^{norm}(\overline{\omega}) = \frac{\sum_\xi J_{esc}(\overline{\omega}, \xi)}{\int \sum_\xi J_{esc}(\overline{\omega}, \xi)\, d\overline{\omega}} = G(\overline{\omega}, R_w) \tag{25}$$

Using (24) and (25), we obtain the following relation for the spectral intensity:

$$J_{esc}(\overline{\omega}) = \sum_\xi J_{esc}(\overline{\omega}, \xi) = \frac{P_{EC}^{tot}}{\pi S_{wall}(1 - R_w)\,\omega_{max}}\, G(\overline{\omega}, R_w), \tag{26}$$

where the function $G(\overline{\omega}, R_w)$ does not depend on the normalized temperature and density profiles (cf. (20)) and turns out to be a weak function of $R_w$.

Here, we illustrate the self-similarity of intensity in the form (25) on the results from the CYNEQ code [20] and extend the analysis of the codes via analyzing the shape of the $P_{EC}(\rho)$ profile for the results mainly from CYNEQ and CYTRAN [21], for temperature and density profiles with the same shape and substantially different volume-averaged values.

## 3. Results

### 3.1. Input Parameters for Analysis of Self-Similarity of ECR Transport

We present the results of analyzing the self-similarity in the following range of input parameters:

- ITER-like geometry parameters: torus major radius, $R_0 = 6.2$ m, minor radius, $a = 2.0$ m, elongation, $k_{elong} = 1.9$, triangularity, $\delta = 0.3$,
- three types of the normalized temperature profile and two types of the normalized density profile, defined by the unified formula:

$$F(\rho) = F_1 + (F_0 - F_1)(1 - \rho^{\beta_F})^{\gamma_F} \tag{27}$$

where $F_0 = F(0)$ and $F_1 = F(1)$ are central and edge values, respectively, and coefficients $\beta_F$ and $\gamma_F$ determine the type of profile (see Tables 1 and 2).
- a wide range of peak values of electron temperature, which includes the following values of the central temperature $T_e(0) = 20$–$55$ keV (with a step of 5 keV) and fixed value of the edge temperature $T_e(1) = 100$ eV (the respective values of volume-averaged temperature are shown in Figure 4 and Tables 1 and 2),
- wide range of reflection coefficient values, from $R_w = 0.6$ to $R_w = 0.9$,
- three types of the profiles of the magnetic field (0D, 1D, and 2D approximations (11)–(13), calculated using the ASTRA code results for plasma equilibrium.

**Table 1.** Parameters of the electron temperature profile (27).

| Edge and Central Values | | Electron Temperature Profile | | | | | |
| | | Parabolic | | Advanced | | ITB | |
| | | $\beta_T = 2.0, \gamma_T = 1.5$ | | $\beta_T = 5.4, \gamma_T = 8.0$ | | $\beta_T = 9.3, \gamma_T = 16.1$ | |
| $T_e(1)$, keV | $T_e(0)$, keV | $<T_e>_V$, keV | $T_e^{eff}$, keV | $<T_e>_V$, keV | $T_e^{eff}$, keV | $<T_e>_V$, keV | $T_e^{eff}$, keV |
| 0.01 | 20 | 8.0 | 11.8 | 8.0 | 12.4 | 10.0 | 14.0 |
| | 25 | 10.0 | 14.7 | 10.0 | 15.5 | 12.5 | 17.5 |
| | 30 | 12.0 | 17.7 | 12.0 | 18.6 | 15.0 | 21.0 |
| | 35 | 14.0 | 20.6 | 14.0 | 21.7 | 17.5 | 24.5 |
| | 40 | 16.0 | 23.6 | 16.0 | 24.8 | 20.0 | 28.0 |
| | 45 | 18.0 | 26.5 | 18.0 | 27.9 | 22.5 | 31.5 |
| | 50 | 20.0 | 29.5 | 20.0 | 31.0 | 25.0 | 35.0 |
| | 55 | 22.0 | 32.4 | 22.0 | 34.1 | 27.4 | 38.5 |

**Table 2.** Parameters of the electron density profile (27).

| | Flat Profile | Non-Flat Profile |
| | $\beta_n = 2.0, \gamma_n = 0.1$ | $\beta_n = 1.5, \gamma_n = 0.5$ |
| --- | --- | --- |
| $n_e(0)$, $10^{20}$ m$^{-3}$ | 1.10 | 1.00 |
| $n_e(1)$, $10^{20}$ m$^{-3}$ | 0.50 | 0.10 |
| $<n_e>_V$, $10^{20}$ m$^{-3}$ | 1.05 | 0.65 |
| $n_e^{eff}$, $10^{20}$ m$^{-3}$ | 1.07 | 0.77 |

We did not analyze the role of polarization scrambling caused by the depolarization of EC waves in their reflections from the wall. It was shown in [38] that for thermonuclear fusion plasmas, polarization scrambling of internal ECR turns out to only weakly influence the net EC radiative power density and the total EC power loss.

Note that in calculations using the CYTRAN code [21], we used its version participated in the benchmarking [22]: (i) the value of critical optical depth in (17) is taken equal to 1.4, according to its definition in the original version of CYTRAN [16]; (ii) the absorption coefficients are taken in the improved form given in Equation (5) in [22]; (iii) magnetic field is taken in the 0D approximation; and (iv) plasma equilibrium parameters (Shafranov shift, elongation, triangularity, and safety factor on the magnetic axis) are taken from ASTRA simulations.

The self-similarity of the normalized EC power density profiles and of the intensity spectra made it reasonable to introduce the following functions: mean normalized

profiles (28) and (29); the relative root-mean-square (rms) deviations (30) and (32) from the profiles (28) and (29), respectively; and the integral characteristics (31) and (33) of the relative rms deviation from the profiles (28) and (29):

$$P_{EC}^{mean}(\rho) = \frac{1}{k_{\max}} \sum_{k=1}^{k_{\max}} P_{EC}^{norm}(\rho, \{T_{e,k}(0) \cdot T_e^{norm}(\rho)\}), \tag{28}$$

$$J_{esc}^{mean}(\overline{\omega}) = \frac{1}{k_{\max}} \sum_{k=1}^{k_{\max}} J_{EC}^{norm}(\overline{\omega}, \{T_{e,k}(0) \cdot T_e^{norm}(\rho)\}), \tag{29}$$

$$\sigma_P(\rho) = \sqrt{\frac{1}{k_{max}} \sum_{k=1}^{k_{max}} \left( \frac{P_{EC}^{norm}(\rho, \{T_{e,k}(\rho)\}) - P_{EC}^{mean}(\rho)}{P_{EC}^{mean}(0)} \right)^2} \tag{30}$$

$$\overline{\sigma}_P = \frac{1}{V_{tot}} \int_0^1 \sigma_P(\rho) \frac{dV}{d\rho} d\rho \tag{31}$$

$$\sigma_J(\overline{\omega}) = \sqrt{\frac{1}{k_{max}} \sum_{k=1}^{k_{max}} \left( \frac{J_{esc}^{norm}(\overline{\omega}, \{T_{e,k}(\rho)\}) - J_{esc}^{mean}(\overline{\omega})}{\max(J_{esc}^{mean}(\overline{\omega}))} \right)^2} \tag{32}$$

$$\overline{\sigma}_J = \int \sigma_J(\overline{\omega}) d\overline{\omega} \tag{33}$$

where the subscript $k$ denotes the temperature profiles with different central temperatures (see Table 1), which corresponds to $k_{max} = 8$.

For the maximum frequency, $\omega_{max}$, defined in Equation (24), the following approximate scaling law was found:

$$\widetilde{\omega}_{max} = \frac{\omega_{max}}{\omega_{B0}} = 0.53 \, T_e^{eff} + 4 \tag{34}$$

$$T_e^{eff} = \int_0^1 T_e(\rho) d\rho \tag{35}$$

The relation (34) is valid for a wide class of the normalized temperature profiles (Figure 5).

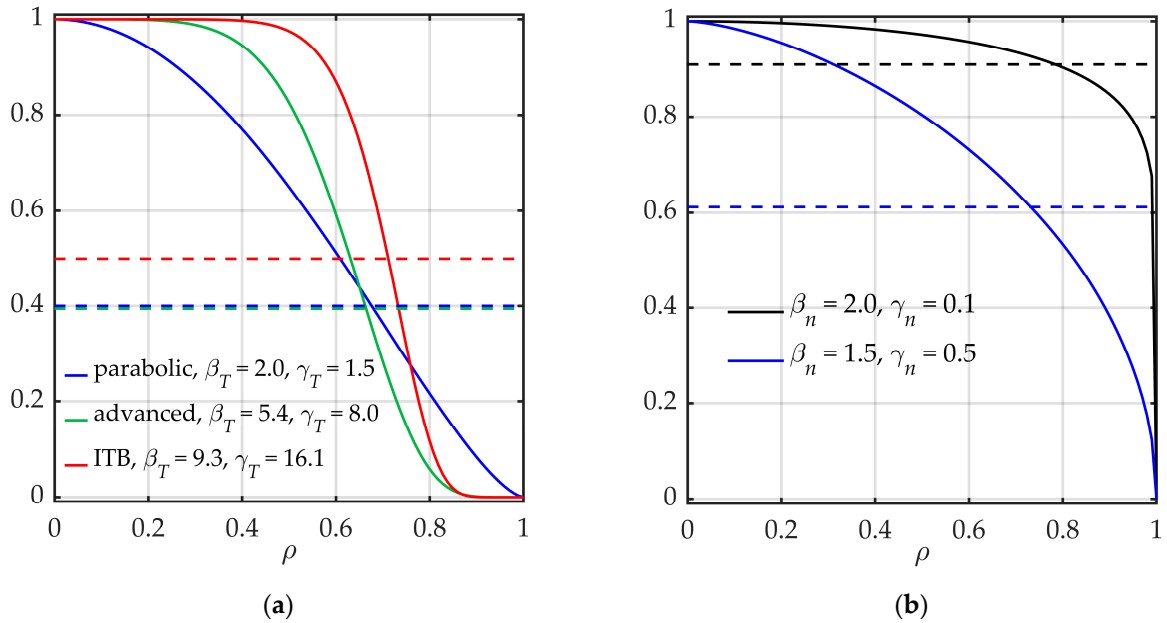

**Figure 4.** Normalized (by maximum value) profiles (27) (solid lines) and corresponding volume averages (dashed lines) for electron temperature (**a**) (see Table 1) and electron density (**b**) (see Table 2).

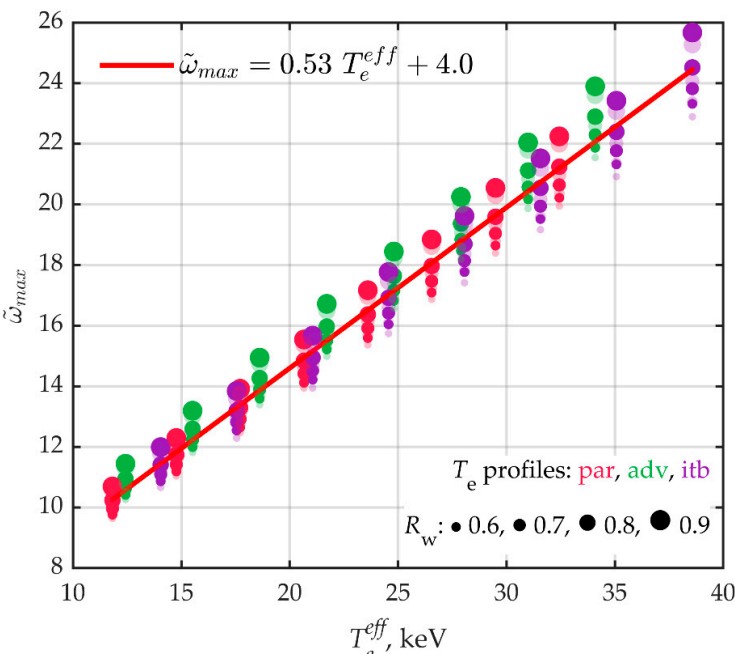

**Figure 5.** Dependence of the maximum frequency defined in (24) (in units of the fundamental cyclotron frequency) on the line-averaged temperature (35) in scenarios with different temperature profiles (see Table 1) and different values of the reflection coefficient. Semitransparent markers correspond to the non-flat density profile, nontransparent markers, to the flat density profile (see Table 2).

### 3.2. Self-Similarity of ECR Transport, Calculated Using CYNEQ and CYTRAN Codes

We start the presentation of results with the calculations using the CYNEQ code for three cases of magnetic field profiles (0D, 1D, and 2D models (11)–(13)) and CYTRAN code with 0D approximation of magnetic field profile. We will illustrate the effect of self-similarity by comparing the results for the absolute values of the functions with their normalized values in the adjacent figures (see Figures 6 and 7). The results are presented for parabolic temperature profile and three values of the central temperature, and a flat profile of the electron density (see Tables 1 and 2 and (27); similar figures for the advanced and ITB temperature profiles can be found in the Supplementary Materials in Figures S2–S5). In the first row, the figures (a) show radial profile of the net EC power loss density (4) and the figures (b) show the corresponding normalized profiles of the net EC power loss density (20). In the second row, the figures (c) present the spectral intensity of the escaping ECR (18) and the figures (d) present the corresponding normalized spectra (25). Figure 6 shows the results for the wall reflection coefficient $R_w = 0.6$, and Figure 7 for $R_w = 0.9$.

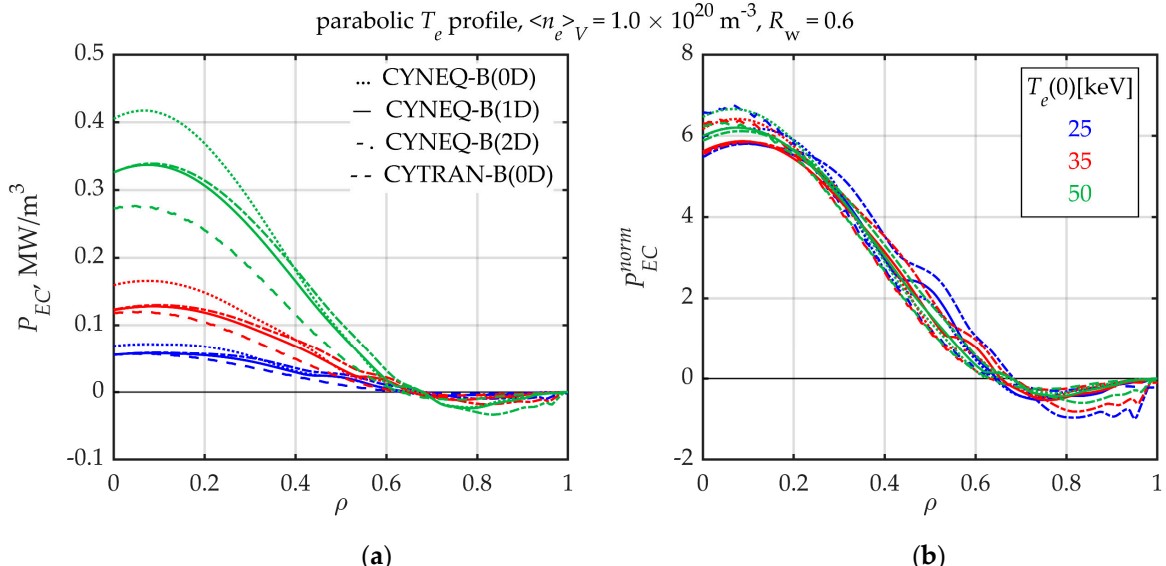

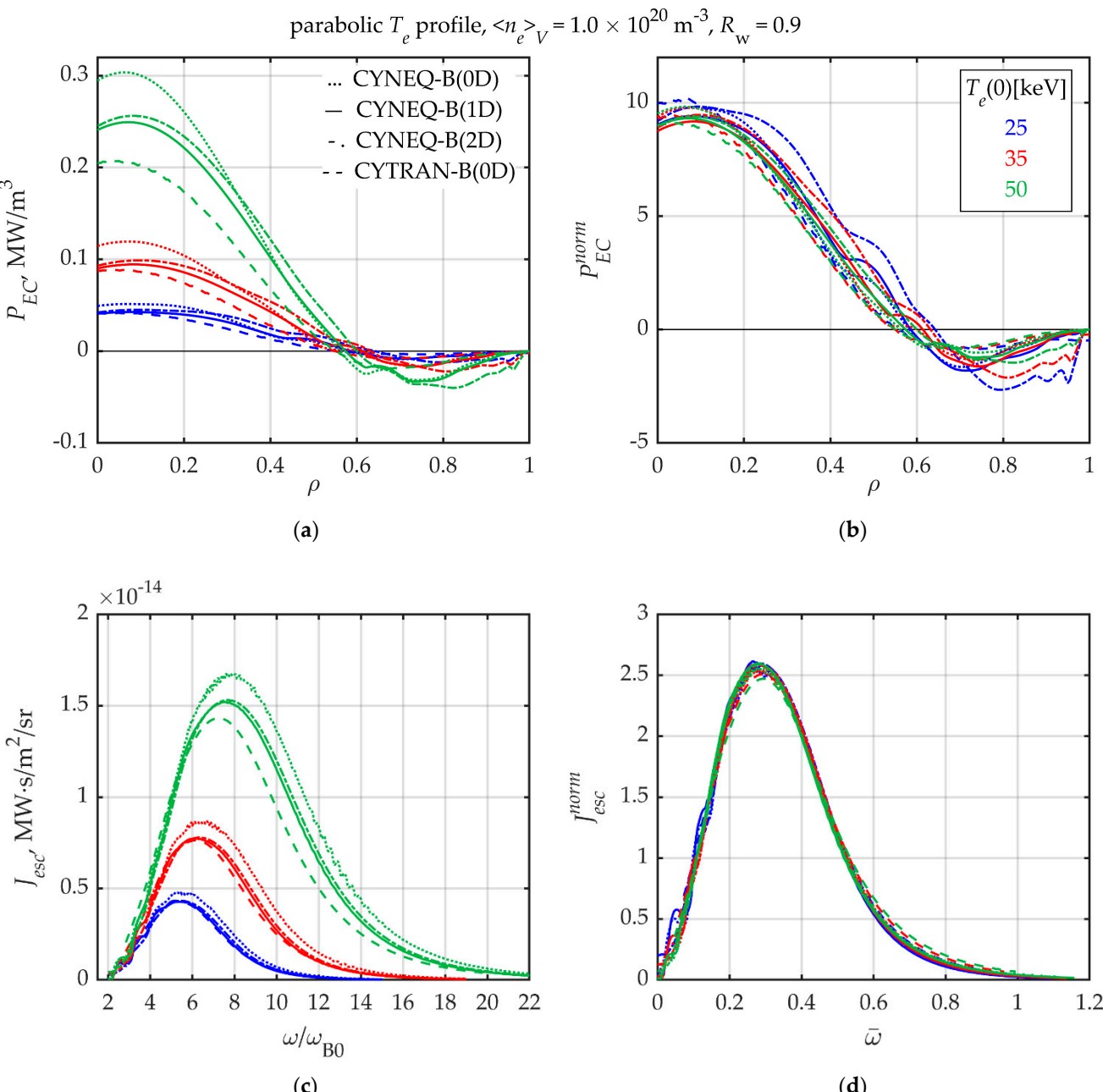

**Figure 7.** The same as in Figure 6, except the wall reflection coefficient is $R_w = 0.9$.

The next step of our analysis of self-similarity (Figures 8–10) for a single profile of electron density (flat profile, see Table 2) is the calculation of the mean profiles (28) of the normalized radial distribution of the net EC power loss density (figures (a)) and the mean spectra (29) of the normalized intensity (figures (b)). Figures 8–10 show the results for three types of the temperature profile (see Table 1 and (27)). It is seen that the degree of self-similarity is high and weakly depends on the reflection coefficient from the wall, $R_w$: the mean profiles for $R_w = 0.6$ and $R_w = 0.9$ are close enough.

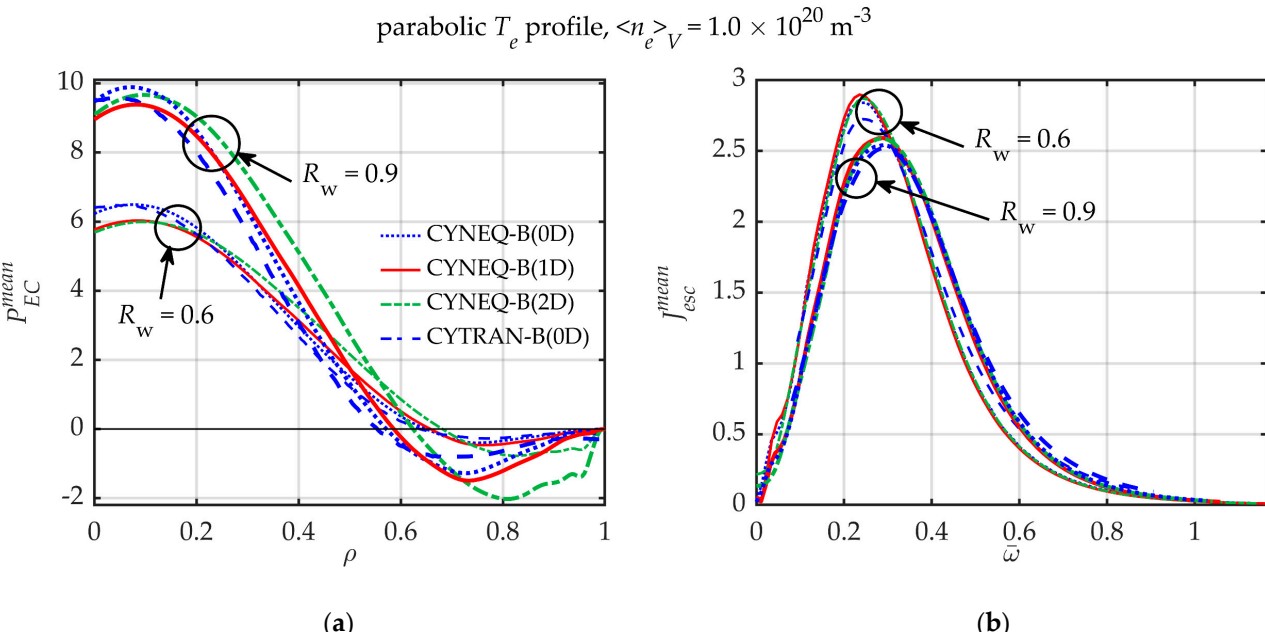

**Figure 8.** Mean profiles (28) of the normalized radial distribution of the net EC power loss density (**a**), and mean spectra (29) of the normalized intensity (**b**), for the parabolic temperature profile and the flat density profile, and $R_w = 0.6$ and $R_w = 0.9$ (see Table 1 and (27)).

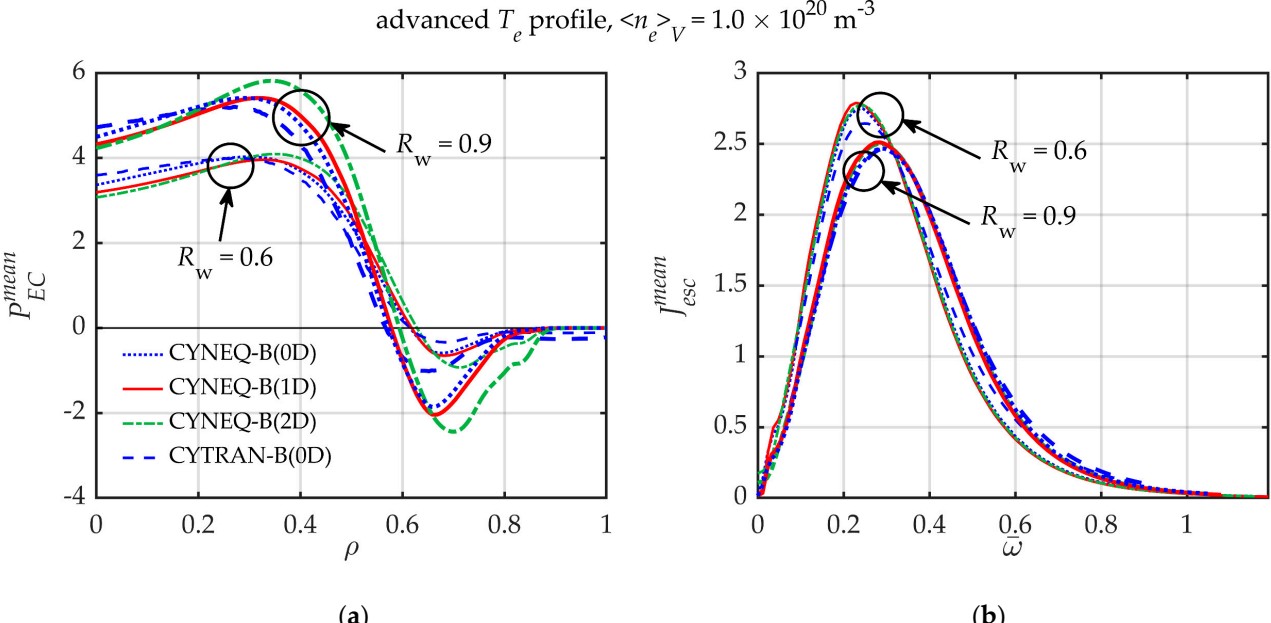

**Figure 9.** The same as in Figure 8, but for the advanced profile of electron temperature (see Table 1 and (27)).

To roughly estimate the contribution of the ECR loss to the power balance, the total (i.e., volume-integrated) ECR power loss can be used. The results for the total power loss in the cases analyzed in Figures 6 and 7 are shown in Table 3. However, it should be noted that for predictive modeling of the tokamak-reactor operation, the spatial distribution of ECR power loss, especially in the central plasma, is of primary interest because of the leading role of central plasma in achieving the conditions of thermonuclear burning.

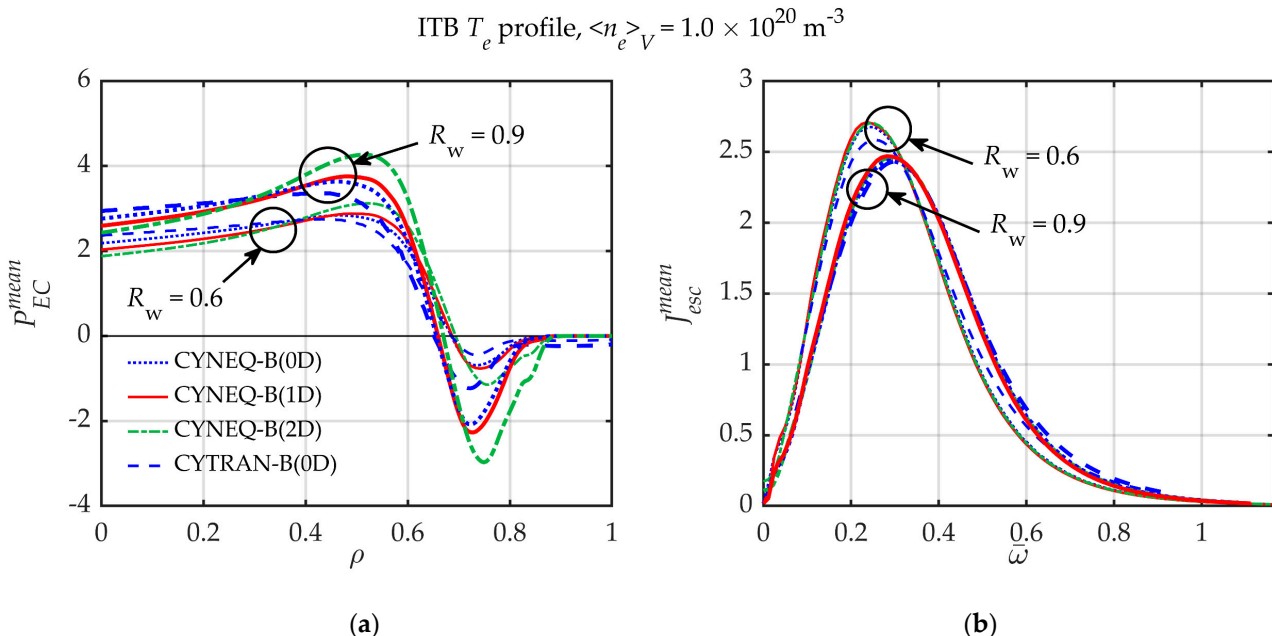

**(a)**　　　　　　　　　　　　　　　　**(b)**

**Figure 10.** The same as in Figure 8, but for the ITB profile of electron temperature (see Table 1 and (27)).

**Table 3.** Total (i.e., volume-integrated) EC power loss for ITER-like tokamak-reactor with plasma parameters from Table 1, calculated using the CYNEQ code, for different approximations of the magnetic field 0D, 1D, and 2D (11)–(13), and CYTRAN code, for constant magnetic field, 0D approximation (13), for different values of wall reflection coefficient, $R_w$.

| $T_e$ profile | $R_w$ | $T_e(0)$, keV | | | | | | | | | | | |
|---|---|---|---|---|---|---|---|---|---|---|---|---|---|
| | | 25 | | | | 35 | | | | 50 | | | |
| | | $P_{EC}$, MW | | | | | | | | | | | |
| | | CYNEQ | | | CYTRAN | CYNEQ | | | CYTRAN | CYNEQ | | | CYTRAN |
| | | 0D | 1D | 2D | 0D | 0D | 1D | 2D | 0D | 0D | 1D | 2D | 0D |
| parabolic | 0.6 | 10.5 | 9.1 | 9.2 | 7.3 | 23.5 | 19.9 | 20.2 | 16.0 | 57.0 | 49.5 | 50.5 | 37.3 |
| | 0.7 | 9.0 | 7.7 | 7.9 | 6.3 | 20.3 | 17.1 | 17.4 | 14.0 | 49.6 | 43.0 | 43.9 | 32.8 |
| | 0.8 | 7.2 | 6.2 | 6.3 | 5.1 | 16.3 | 13.8 | 14.0 | 11.5 | 40.5 | 35.1 | 35.8 | 27.3 |
| | 0.9 | 4.8 | 4.1 | 4.2 | 3.5 | 11.1 | 9.4 | 9.5 | 8.0 | 28.1 | 24.4 | 24.8 | 19.4 |
| advanced | 0.6 | 14.9 | 13.0 | 13.4 | 10.1 | 33.8 | 28.8 | 29.7 | 22.9 | 83.3 | 72.6 | 74.8 | 54.8 |
| | 0.7 | 12.7 | 11.1 | 11.5 | 8.8 | 29.1 | 24.8 | 25.6 | 20.0 | 72.5 | 63.2 | 65.1 | 48.3 |
| | 0.8 | 10.2 | 8.9 | 9.2 | 7.2 | 23.4 | 20.0 | 20.7 | 16.5 | 59.2 | 51.6 | 53.2 | 40.2 |
| | 0.9 | 6.8 | 5.9 | 6.1 | 4.9 | 15.9 | 13.6 | 14.1 | 11.5 | 41.3 | 35.9 | 37.1 | 28.7 |
| ITB | 0.6 | 18.6 | 16.7 | 17.6 | 12.2 | 42.7 | 37.5 | 39.4 | 27.8 | 106.5 | 95.5 | 99.7 | 67.6 |
| | 0.7 | 15.9 | 14.2 | 15.0 | 10.6 | 36.7 | 32.2 | 33.9 | 24.3 | 92.3 | 82.8 | 86.6 | 59.6 |
| | 0.8 | 12.6 | 11.3 | 12.0 | 8.6 | 29.4 | 25.8 | 27.2 | 19.9 | 75.1 | 67.4 | 70.5 | 49.4 |
| | 0.9 | 8.4 | 7.5 | 8.0 | 5.9 | 19.9 | 17.5 | 18.5 | 13.9 | 52.0 | 46.7 | 48.9 | 35.2 |

The final step of our analysis for a flat profile of electron density is the calculation of the root-mean-square (rms) deviations (30)–(33) for three types of electron temperature profile (see Table 1).

Figure 11 shows spatial profiles of the relative deviation (30) of the normalized profiles of the EC power loss density from the mean normalized profile (28) (figure (a)), and of the spectral profiles of the relative deviation (32) of the normalized spectral intensity of the escaping ECR from the mean normalized profile (29) (figure (b)) for parabolic temperature profile (similar figures for the advanced and ITB temperature profiles can be found in the Supplementary Materials in Figures S6 and S7). The results show, respectively, spatial and spectral functions for different values of the wall reflection coefficient $R_w$.

Figure 12 shows volume-averaged deviation (31) of the normalized profile of the net EC power loss density from the mean normalized profile (28) (figure (a)), and spectrum-averaged deviation (33) of the normalized spectra of EC radiation intensity from the mean normalized spectrum (29) (figure (b)) as a function of $R_w$ for parabolic temperature profile (similar figures for the advanced and ITB temperature profiles can be found in the Supplementary Materials in Figures S8 and S9).

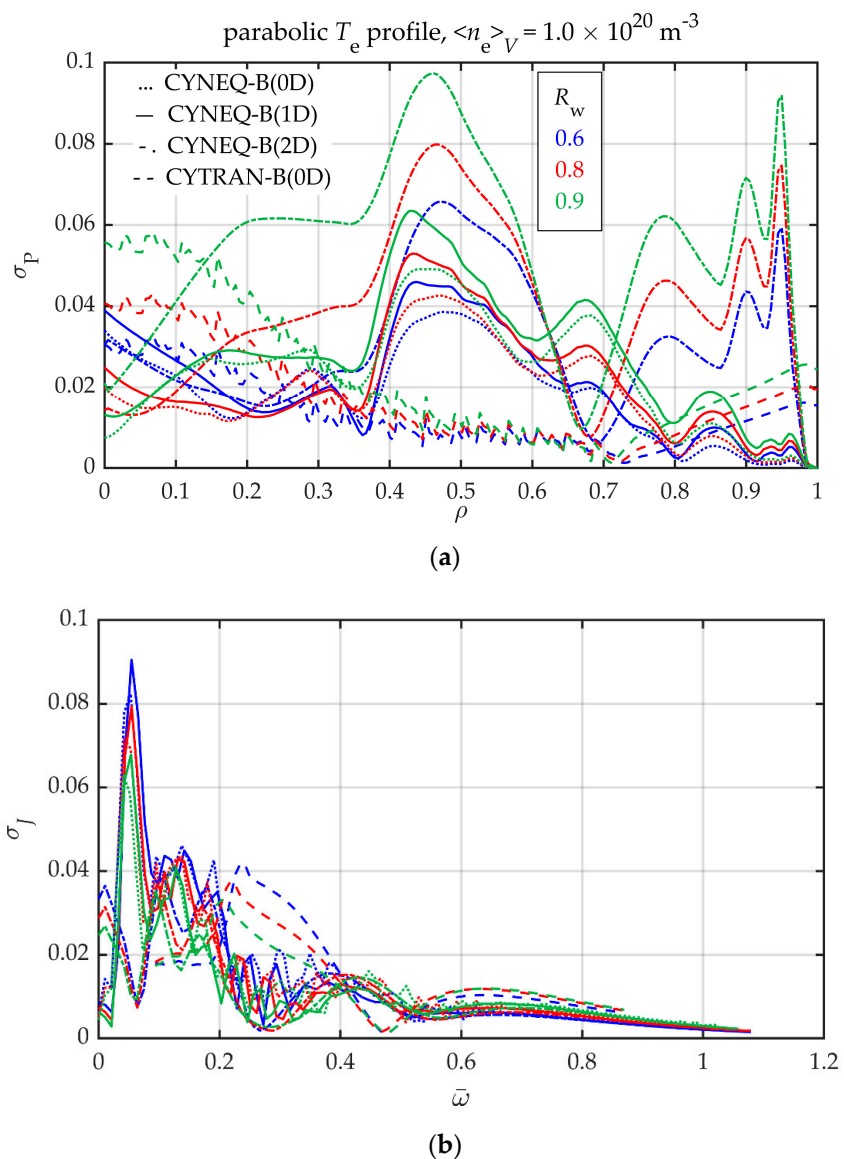

**Figure 11.** (**a**) Profiles of the relative deviation (30) of the normalized profiles of the EC power loss density from the mean normalized profile (28). (**b**) Profiles of the relative deviation (32) of the normalized spectral intensity of the escaping ECR from the mean normalized spectrum (29) for the parabolic profile of $T_e$ and the flat profile of $n_e$ (see Table 1) and different values of $R_w$. The color of the curve corresponds to the $R_w$ value in the inset, the line type of the curve corresponds to the code shown in the inset.

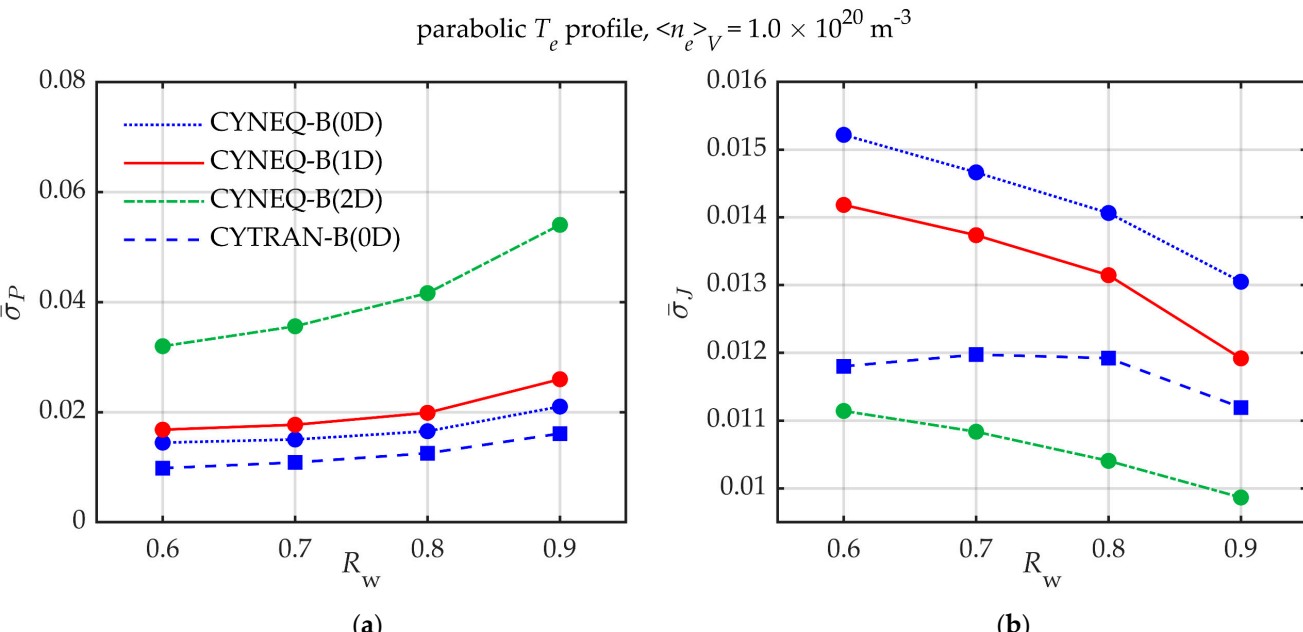

**Figure 12.** Volume-averaged deviation (31) of the normalized profile of the net EC power loss density from the mean normalized profile (28) (**a**), and spectrum-averaged deviation (33) of the normalized spectra of EC radiation intensity from the mean normalized spectrum (29) (**b**), as a function of the wall reflection coefficient for the parabolic profile of the electron temperature and the flat profile of electron density (see Table 1 and (27)).

We now turn to the analysis of the dependence of self-similarity on the electron density profile to complement the previous analysis for a flat profile of the electron density (see Table 2). Figure 13 shows the comparison of the accuracy of the self-similarity for the flat and non-flat profiles of the electron density for parabolic temperature profile and different central temperatures in the range 20–55 keV (see Tables 1 and 2 and (27); similar figures for the advanced and ITB temperature profiles can be found in the Supplementary Materials in Figures S10 and S11). Figure (a) shows a comparison of the mean profiles (28) of the normalized radial distribution of the net EC power loss density, and figure (b) shows comparison of the mean spectra (29) of the normalized intensity of EC radiation, for the wall reflection coefficient $R_w = 0.9$. The volume-averaged deviation (30) of the normalized profile of the net EC power loss density from the mean normalized profile are shown as a function of $R_w$ in figures (c), and the same dependence of the spectrum-averaged deviation (32) of the normalized spectra of EC radiation intensity from the mean normalized spectrum is shown in figures (d). It is seen from Figure 13 that the dependence of the accuracy of the self-similarity on the type of the density profile is weak.

The results of analyzing the self-similarity in the case of the non-flat profile of electron density (see Table 2 and (27)) are presented in the Supplementary Materials (Figures S12–S27).

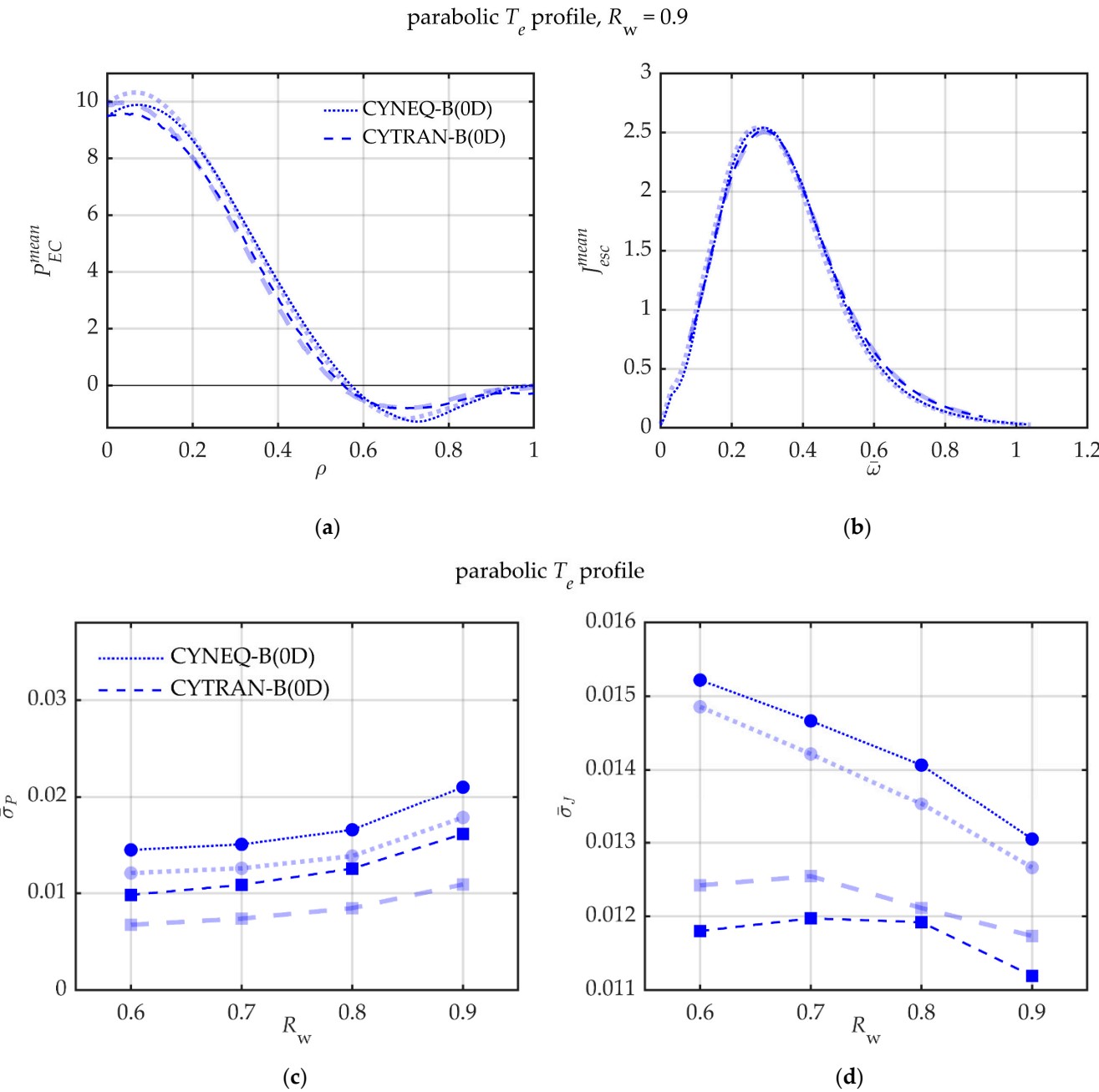

**Figure 13.** Comparison of the accuracy of the self-similarity for the flat (non-transparent thin lines) and non-flat (transparent thick lines) electron density profile, parabolic temperature profile with different central temperatures in the range 20–55 keV (see Tables 1 and 2 and (27)). (**a**) Mean profiles (28) of the normalized radial distribution of the net EC power loss density; (**b**) mean spectra (29) of the normalized intensity of EC radiation; (**c**) deviation (30) of the normalized profile of the net EC power loss density from the mean normalized profile; and (**d**) deviation (32) of the normalized spectra of EC radiation intensity from the mean spectrum.

### 3.3. Self-Similarity of ECR Transport, Calculated Using CYNEQ, RAYTEC, and EXACTEC Codes

The analysis of Section 3.2 based on the results of our calculations using the CYNEQ and CYTRAN codes is extended here to similar calculations in the cases for which we have the results from the RAYTEC and EXACTEC codes. A comparison, similar to that in Figure 6, with the latest results [35] from the RAYTEC code is shown in Figures 14–16. Figures 14 and 15 use the results for five cases considered in [35]. Figure 16 analyzes the effect of the plasma equilibrium (Shafranov shift of the plasma column towards a weaker

magnetic field due to plasma diamagnetism) on the power loss density profile. Note that the CYNEQ code calculations take into account the plasma equilibrium effects, while the RAYTEC code takes into account the inhomogeneity of the toroidal magnetic field and does not take into account the effects of plasma equilibrium.

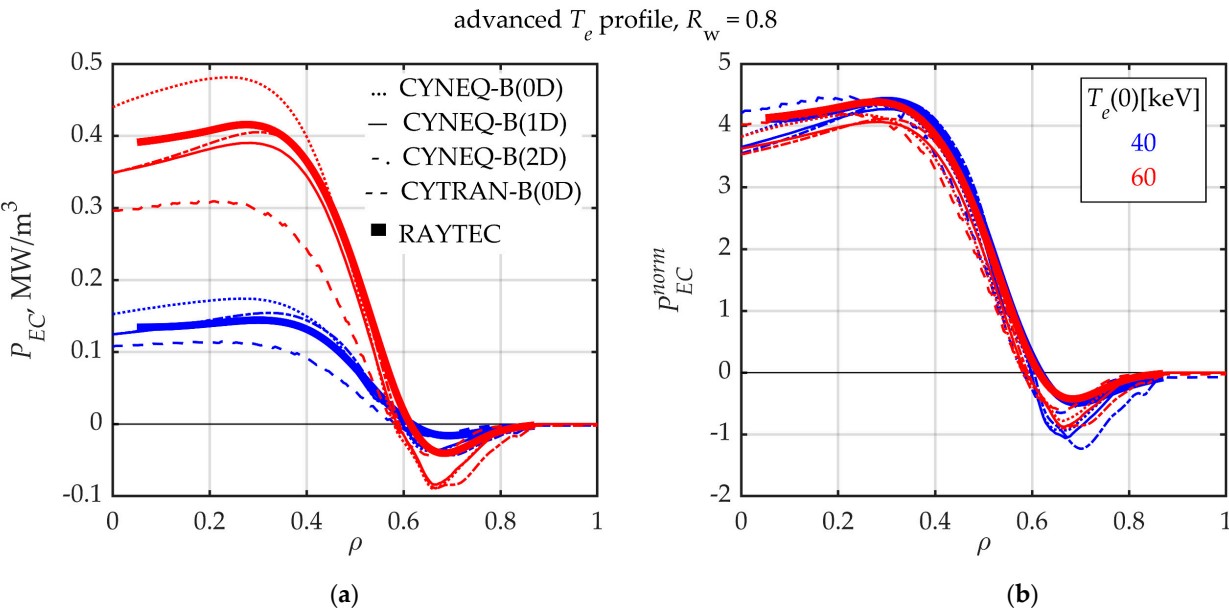

(**a**)           (**b**)

**Figure 14.** Self-similarity of ECR transport, calculated using the CYNEQ, CYTRAN, and RAYTEC codes, for tokamak-reactor conditions with the advanced profile of electron temperature, different values of the central temperature indicated in the inset, and a peaked profile of the electron density (see Table 1 and (27) in this paper, and case B and C in Table 1 and Figure 2 in [35]), wall reflection coefficient $R_w = 0.8$, $R_0 = 8.5$ m, a = 2.7 m, $k_{elong} = 1.7$, and $B_0 = 6$ T: (**a**) radial profile of the net EC power loss density (4) (the color of the curve corresponds to the central temperature indicated in the inset) and (**b**) the corresponding normalized profiles of the net EC power loss density (20). CYNEQ code takes into account plasma equilibrium calculated using the ASTRA code for $I_p = 20$ MA.

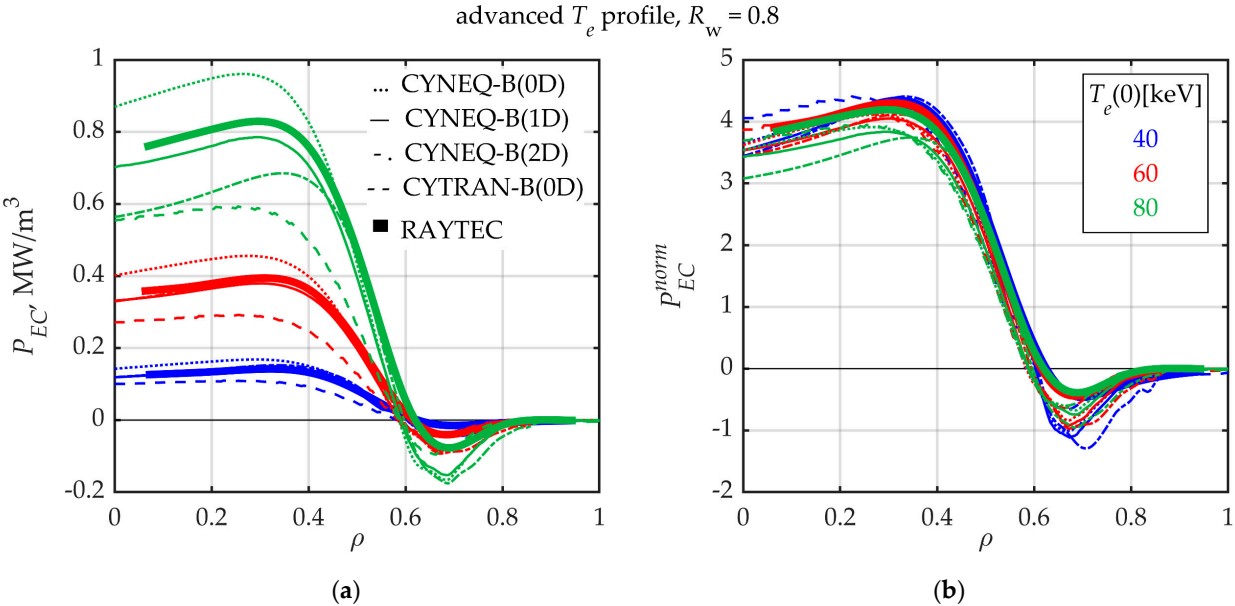

(**a**)           (**b**)

**Figure 15.** The same as in Figure 14, but for flat density profile, there are different central temperatures (indicated in the inset) and different central electron densities (see case A in Table 1 and Figure 3 in [35]).

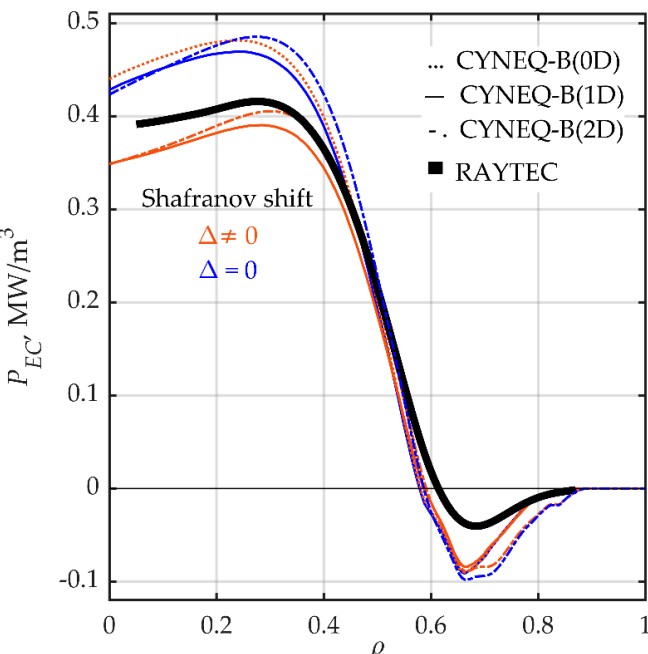

**Figure 16.** Radial profile of the net EC power loss density (4), calculated using the CYNEQ and RAYTEC codes for tokamak-reactor conditions with the advanced profile of electron temperature and the peaked electron density profile (see case B in Table 1 and Figure 2 in [35]), $R_w = 0.8$, $R_0 = 8.5$ m, $a = 2.7$ m, $k_{\text{elong}} = 1.7$, $B_0 = 6$ T, and $I_p = 20$ MA. CYNEQ calculations for three approximations of the magnetic field (0D, 1D, and 2D (11)–(13)) were carried out with account of plasma equilibrium (nonzero Shafranov shift, $\Delta$, brown lines) and with neglect of equilibrium effects (zero Shafranov shift, blue lines).

A comparison with EXACTEC code, similar to Figures 14–16 for RAYTEC code, is shown in Figures 17–19. Figures 17 and 18 use the results for two cases of central electron temperature and two types of temperature profile considered in [22]. Figure 19 analyzes the effect of the plasma equilibrium (Shafranov shift) on the power loss density profile. Note that the EXACTEC code calculates the results for the case of a plasma in a straight circular cylinder with a uniform axial magnetic field, where there are no plasma equilibrium effects, unlike CYNEQ calculations, where plasma equilibrium is taken into account.

The last step in analyzing the self-similarity of the ECR transport in toroidal plasmas with highly reflecting walls is a comparison of the CYNEQ results with the simple analytic model known as the locally applied Trubnikov formula (LATF) (see Appendix in [25]). This formula is based on an intuitive application of the formula suggested in [8] as a fit of the numerical results [5] for the total (i.e., volume-integrated, not spatially resolved) ECR power loss of a homogeneous plasma in a constant magnetic field. Formula [8] included the effect of wall reflection and inhomogeneity of the toroidal magnetic field. However, this formula was not intended to be used to estimate the spatial profile of the ECR power loss density. An intuitive extension of the formula [8] gives reasonable results: good results for the total power loss in an inhomogeneous plasma and less accurate results for the spatial profile (for more details see [25]). According to the main assumption of the LATF, this formula, in principle, cannot describe the inversion of the sign of $P_{\text{EC}}(\rho)$ in the peripheral plasma. The main features of LATF are illustrated here in Figure 20, which compares the results from Figure 6 for the CYNEQ code with LATF. It is seen that the self-similarity that is ad hoc assumed in an analytic model, does not guarantee satisfactory accuracy of the results. The latter is especially important in the central hot plasma, where LATF significantly underestimates the ECR power loss density.

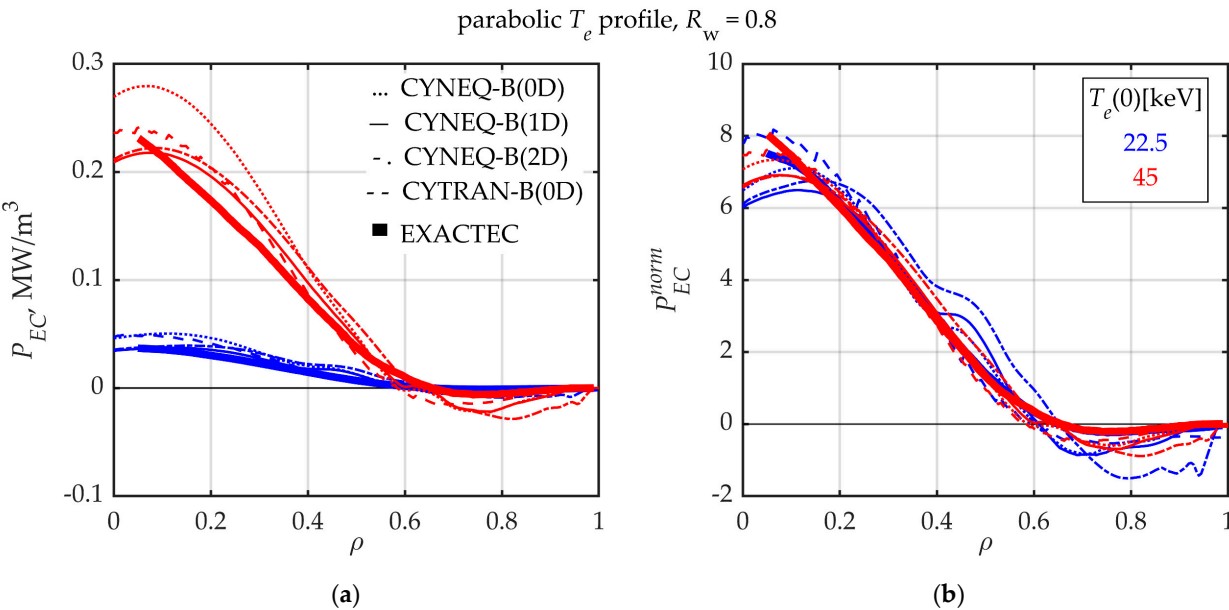

**Figure 17.** Self-similarity of ECR power loss density profiles, calculated using the CYNEQ, CYTRAN, and EXACTEC codes, for tokamak-reactor conditions with the parabolic profile of electron temperature, different values of the central temperature indicated in the inset, and a flat profile of the electron density (see Tables 1 and 2 and (27) in this paper, and the graphs on the left side of Figures 6 and 7 in [22]), wall reflection coefficient $R_w = 0.8$, $R_0 = 6.2$ m, a = 2.0 m, $k_{elong} = 1.0$, and $B_0 = 5.3$ T: (**a**) radial profile of the net EC power loss density (4) (the color of the curve corresponds to the central temperature indicated in the inset); (**b**) the corresponding normalized profiles of the net EC power loss density (20). CYNEQ code takes into account plasma equilibrium calculated using the ASTRA code for $I_p = 10$ MA, $T_e(0) = 22.5$ keV and $I_p = 15$ MA, and $T_e(0) = 45$ keV.

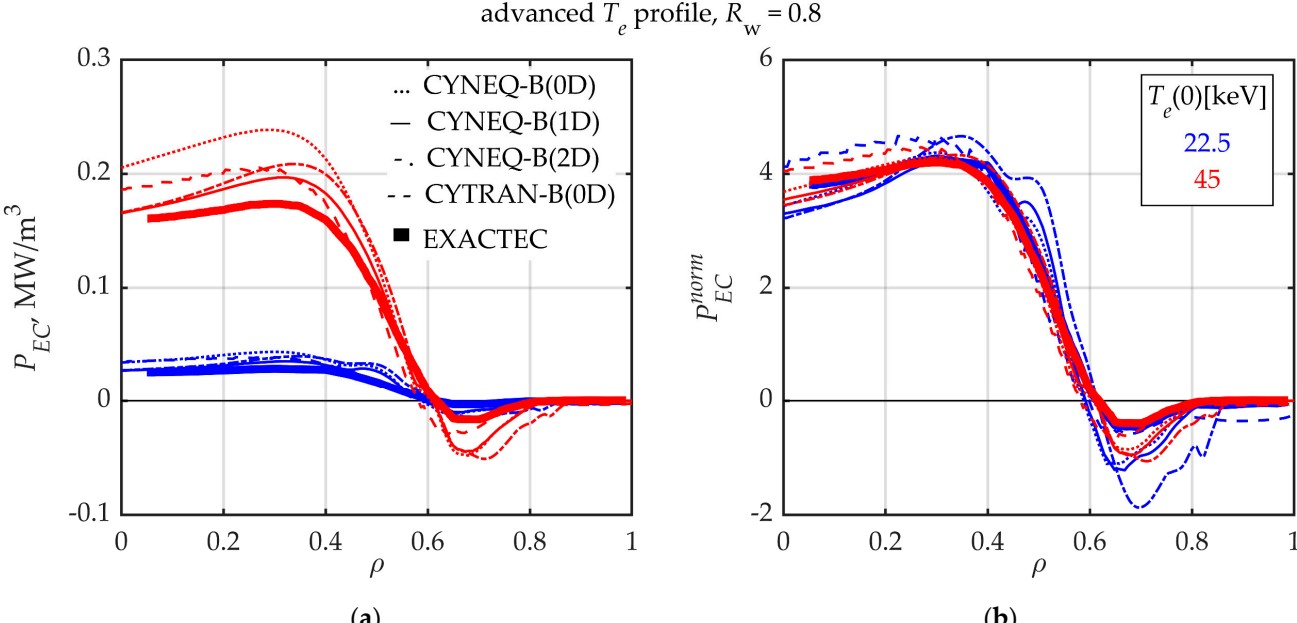

**Figure 18.** The same as in Figure 17 but for the advanced electron temperature profile.

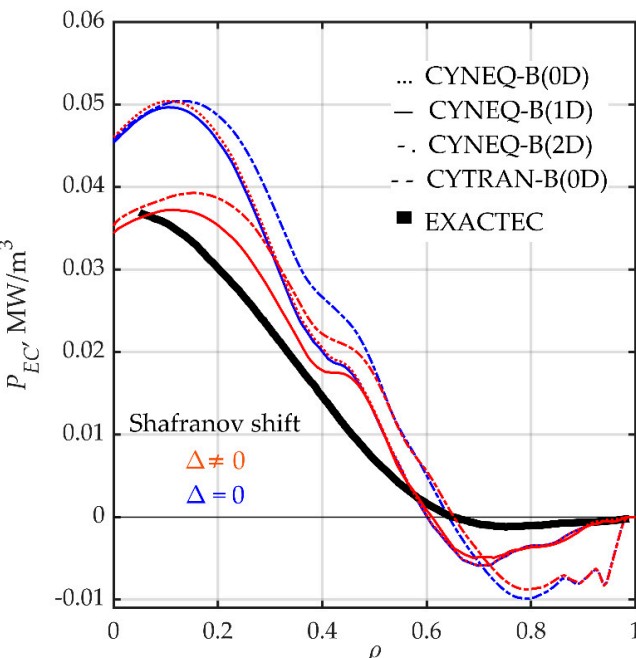

**Figure 19.** Radial profile of the net EC power loss density (4), calculated using the CYNEQ and EXACTEC codes for tokamak-reactor conditions with the parabolic profile of electron temperature, $T_e(0) = 22.5$ keV, and the flat electron density profile (see plots on the left side of Figure 6 in [22]), $R_w = 0.8$, $R_0 = 6.2$ m, a = 2.0 m, $k_{elong} = 1.0$, $B_0 = 5.3$, and $I_p = 10$ MA. CYNEQ calculations for three approximation of the magnetic field (0D, 1D, and 2D (11)–(13)) were carried out with account of plasma equilibrium (nonzero Shafranov shift, $\Delta$, brown lines) and with neglect of equilibrium effects (zero Shafranov shift, blue lines).

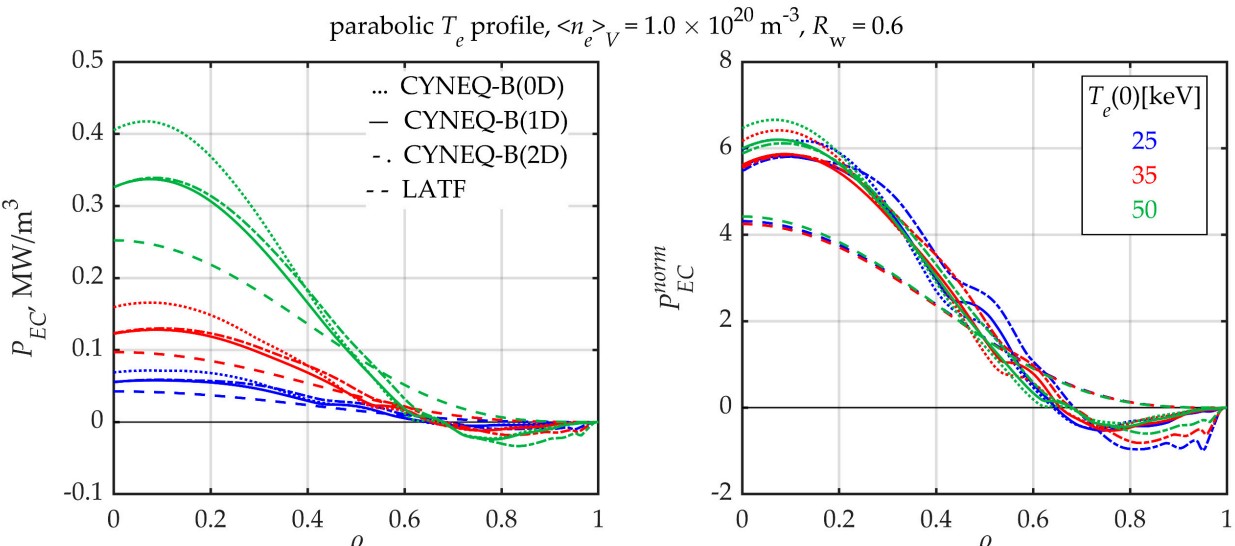

**Figure 20.** The same as in Figure 6 but the CYNEQ results are compared with the locally applied Trubnikov formula (LATF) [25].

### 3.4. Analysis of Self-Similarity of Continuous-Spectrum Transport for Model Transport Coefficients

The analysis of self-similarity of continuous-spectrum radiation transfer in plasmas with highly reflecting walls, carried out in the case of ECR transport, can be extended to show weak dependence of the phenomenon on the parameters of the problem. Now we

consider the model case of the emission and absorption coefficients, different from the ECR case. We choose the following type of the absorption coefficient:

$$\kappa(\rho, \omega) = \kappa_0 \cdot \frac{n_e(\rho)}{n_e(0)} \exp\left(-\frac{(\omega - \omega_{min})^2}{\omega_D^2}\right),$$

$$\omega_D = \omega_{min} \sqrt{\frac{T_e(\rho)}{T_e(0)}}, \tag{36}$$

where the half-width of the spectrum, $\omega_D$, depends on the temperature, $T_e(\rho)$, similarly to Doppler broadening. The emissivity is related to the absorption coefficient in the same way, according to Kirchhoff's law at low frequencies. All other input parameters are taken the same as in the case of ECR transport. The characteristic value of the absorption coefficient, $\kappa_0$, which determines the optical thickness, is also taken in the same range, because our goal is to analyze the sensitivity of the self-similarity of radiative transfer to spectral dependence of emissivity and absorption. The results for the spatial profile of radiation loss and radiation intensity are shown below in Figure 21.

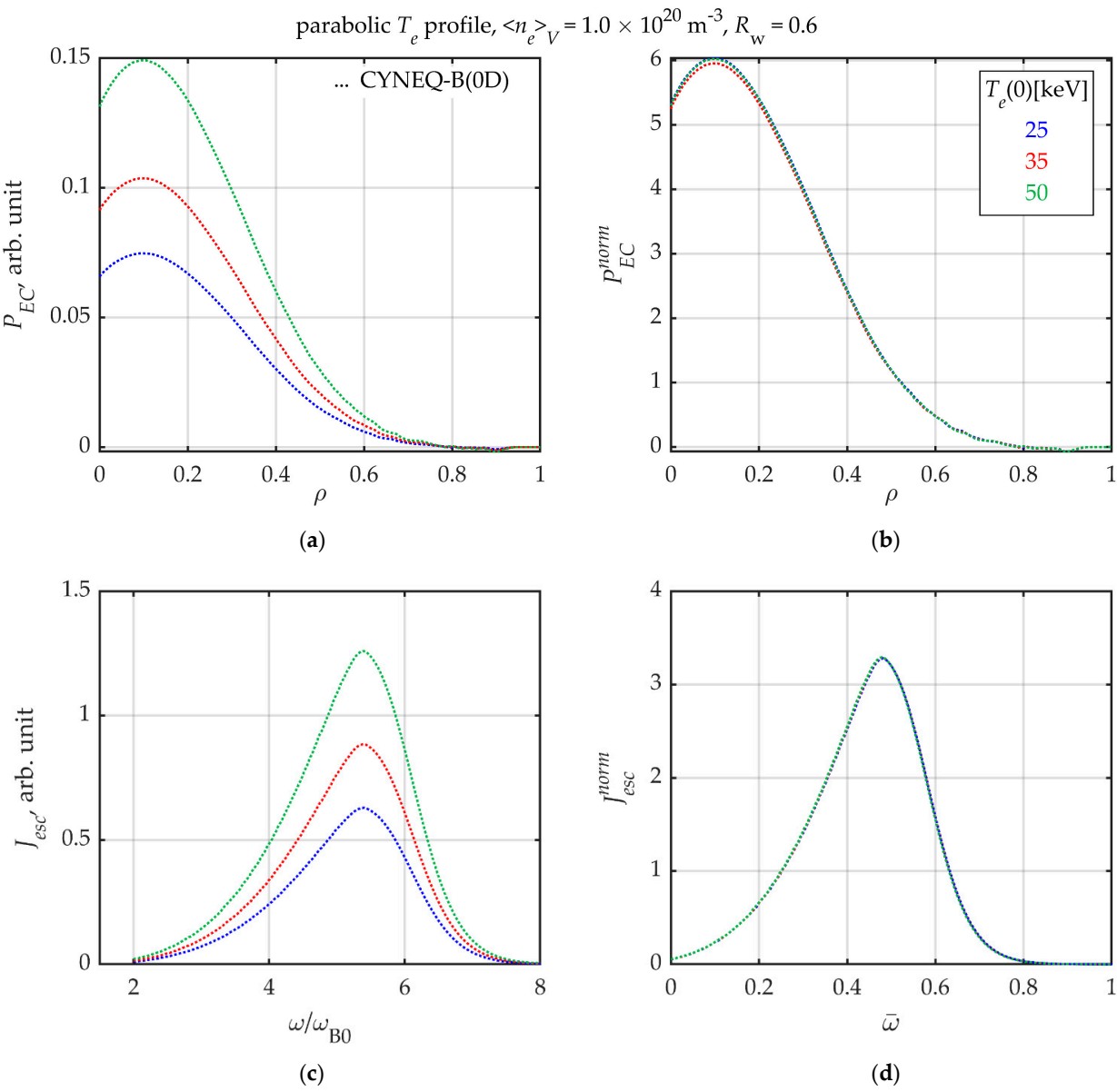

**Figure 21.** The same as in Figure 6 but for the absorption coefficient (36).

More results for the model considered in this subsection can be found in the Supplementary Materials (Figures S28–S30). These results strengthen the conclusion based on the results for the ECR transport: the self-similarity of the main characteristics of the radiative transfer in the continuous spectrum in plasmas with highly reflecting walls; namely, the power balance profile and the radiation intensity spectrum, is observed in a wide range of parameters of the problem.

## 4. Discussion and Conclusions

The similarity of the radiative transfer (RT) in continuous spectrum in plasmas with highly reflecting walls is shown here for the main characteristics of RT, namely the spectral distribution of the radiation intensity escaping from a toroidal plasma, $J_{esc}(\omega)$, defined in (18), and the spatial distribution of the net power loss $P_{EC}(\rho)$, defined in (4). The results are presented for the transport of electron cyclotron radiation (ECR) in hot Maxwellian plasmas under conditions of a thermonuclear fusion tokamak-reactor and for RT with model functions of absorption and emission of radiation, connected by Kirchhoff's law at low frequencies. The revealed similarity of RT is a consequence of the symmetry of highly reflecting walls (namely, toroidal symmetry of vacuum chamber in tokamaks, including the ITER tokamak [2] (under construction) and various next-step projects called DEMO [3]), which predetermines the isotropization of the radiation intensity with respect to angular and spatial variables in some parts of the phase space of the problem.

The use of symmetry-based properties of RT made it possible to create the fast-routine algorithm [16] ("simulator code" that is comparable to simple analytical expressions in terms of computation speed, as opposed to very time-consuming full transport calculations), which is used in the numerical modeling of the spatial profile of ECR power loss in tokamak-reactors. For example, numerical codes CYTRAN [16,21] and EXACTEC [23] were used in predictive modeling of operation scenarios, respectively, for ITER [11] and DEMO [3]. For CYNEQ code [20], these examples include predictive modeling [12] of quasi-steady-state scenarios of ITER operation, analysis of quenching of temperature rise by ECR loss for ITER long pulse operation [15], and elaboration of new ECR-based diagnostics for DEMO. The latter includes the diagnostics of superthermal electrons from ECR spectrum at high harmonics of fundamental EC frequency [39] and the complex diagnostics of electron temperature in central plasma, based on the high-harmonic ECR measurements and the Thomson scattering [40].

The problem of coupling for tokamak-reactors the codes for the ECR transport at moderate and high harmonics of EC fundamental frequency ($n \geq 3$), such as SNECTR [17], CYTRAN, CYNEQ, EXACTEC, and RAYTEC [25], with the codes for EC resonance heating (ECRH) and EC current drive (ECCD) at low harmonics ($n = 1$ and $n = 2$), such as codes participated in the benchmarking of ECRH and ECCD in ITER [41], was discussed in [42] with respect to a coupling of CYNEQ with the following codes for ECRH and ECCD: the GENRAY code [43] for ray trajectory propagation of EC waves for given electron velocity distribution function (eVDF) and the CQL3D code [44] for space–time evolution of eVDF. The necessity of such coupling is motivated by a strong effect of the deviation of eVDF from Maxwellian, in the form of superthermal electrons, on the spatial profile of ECR power loss. For the model eVDFs and ITER-like condition, this effect was estimated using CYNEQ [10] and RAYTEC [45]. The simulations [46,47] of VDF of superthermal electrons using the ray tracing code TORBEAM [48] and the kinetic code RELAX [49] made it possible to evaluate in [46,50] the effect of superthermals on the ECR power profile under conditions of a strong ECRH and ECCD for ITER-like plasma. These studies continued the investigations of the impact of superthermal electrons on the ECR power loss in [51,52] in the case of various model eVDFs. The first step in self-consistent description of the wave transport and kinetics of electrons was made in analytic models [18,53], and the respective estimates [54] for ITER-like conditions have shown the effect of the transport of internal (not injected) ECR on the deviation of eVDF from Maxwellian.

The symmetry-based effect of ECR intensity isotropization in tokamaks was also used in the following works. Application of the CYNEQ code to the problem [55,56] of the start-up stage of tokamak-reactor operation made it possible to create a model [57] of multi-pass absorption of external EC radiation at initial stage of discharge in ITER. Application of ECR intensity isotropization to the problem of interpreting the ECR spectra at down-shifted frequencies (see, e.g., [58]) made it possible to reconstruct the VDF of superthermal electron in the outer layer of plasma in tokamak T-10 [59].

Let us discuss the relationship between the revealed self-similarity and the accuracy of numeric codes.

(i) Self-similarity was found using the results of mass calculations using the fast-routine codes CYNEQ [18–20] and CYTRAN code [16,21] (in its version participated in the benchmarking [22], see Section 3), which have been shown to have good accuracy when compared in [12,22,24] with the codes SNECTR [17] and RAYTEC [25] based on ab initio modeling. A comparison of CYNEQ and CYTRAN results with the latest results from RAYTEC [35] is made in Section 3.3. Thus, the self-similarity phenomenon is supported by the reliability of the CYNEQ and CYTRAN results.

(ii) The degree of self-similarity revealed in a wide range of parameters of the RT problem shows that some success of very simple models for the spatial profile of power loss $P_{EC}(\rho)$, such as the LATF formula [25], is not accidental. However, the similarity that is ad hoc assumed in an analytic model, does not guarantee satisfactory accuracy of the results (e.g., LATF, in principle, cannot describe the inversion of the sign of $P_{EC}(\rho)$ in the peripheral plasma). Consequently, self-similarity alone cannot be a measure of the accuracy of numeric codes.

(iii) The revealed self-similarity opens up new possibilities for constructing simple analytic models for $P_{EC}(\rho)$ and $J_{esc}(\omega)$, which could be used in various multi-parametric analyses of a complex problem, where the simplest versions of various components are required. For example, the self-similar profiles of $P_{EC}(\rho)$ combined with approximate formulas for total (i.e., volume-integrated) EC power loss can be used to assess the contribution of the ECR power loss to the local electron power balance in tokamak-reactors. The self-similar profiles of $J_{esc}(\omega)$ can be used in the ongoing analysis of the role of the internal ECR, escaping from the magnetized plasma in thermonuclear fusion facilities, as a source of thermal and electromagnetic loads on in-chamber components and diagnostic tools (see [13] for ITER).

**Supplementary Materials:** The following are available online at https://www.mdpi.com/article/10.3390/sym13071303/s1.

**Author Contributions:** The authors equally contributed to this work. All authors have read and agreed to the published version of the manuscript.

**Funding:** This research received no external funding.

**Institutional Review Board Statement:** Not applicable.

**Informed Consent Statement:** Not applicable.

**Data Availability Statement:** Not applicable.

**Acknowledgments:** The authors are grateful to A.Y. Dnestrovskii for consulting on the plasma equilibrium in the ASTRA code, and A.R. Polevoi for collaboration in applying the CYNEQ code to the modeling of ITER operation scenarios.

**Conflicts of Interest:** The authors declare no conflict of interest.

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
