# Peer review of "Self-Similarity of Continuous-Spectrum Radiative Transfer in Plasmas with Highly Reflecting Walls"

_symmetry, doi:10.3390/sym13071303_

Round 1

Reviewer 1 Report

In this work, the authors compared the EC power and intensity with different simulation codes, electron temperature, and other parameters. The main results are the self-similarity of power and intensity at different conditions. Since this work is purely a numerical simulation, the similarity introduced in the manuscript is purely a result of mathematics and can be tracked in the equations.

In Eq. 16, q is proportional to Te. Therefore, Jesc is also proportional to Te in Eq. 18. Also, JBB is proportional to Te in Eq. 6. Therefore, J is proportional to Te in Eq. 19. Similarly, in Eq. 5, the power is also proportional to Te. These equations make me surprise that the authors will finally achieve a similar pattern of normalized P and J w.r.t Te. It looks like the final results do not depend on the χ in Eq. 15, although the authors didn’t show the exact math expression of it.

There is no major issue found in this work. I would like to suggest a minor revision for the authors to add the explanation of χ, and double-check the formation of some variables: many of them are not italic.

Reviewer 2 Report

Radiative transfer in plasmas is an important topic for a wide class of plasma problems. The paper describes a radiative transfer model (reduced to a single transport equation for the intensity) dependent on the macroscopic parameters of the plasma. Semi-analytic description is presented in case of the geometric symmetry of the bounding walls enhanced by the diffuse reflectivity. Such simplified model could be useful for a range of plasma parameters and wall reflectivities and hence the paper could be published after minor revision.

A few minor comments:

- There are different approximations used for the right hand side term of radiation transfer equation. E.g. equation (1). 
Often in the lab plasma e.g.
Hizanidis et al, Physics of Plasmas 17, 022505 (2010)
and in astro plasma 
Bian et al, The Astrophysical Journal,  873, 33 (2019)
the Fokker-Planck equation is used to describe radio-wave scattering.
The comparison to these approximations would be useful to understand the advantages and disadvantages.

- it is stated that self-similarity opens up new possibilities for constructing simple analytic models. Please elaborate and exemplify this assertion. 

- page 37 states that "The use of symmetry-based properties of RT made it possible to create the fast-routine algorithms...". Can the authors specify what means "fast", and instead quantitatively state how fast?

Reviewer 3 Report

Dear Authors, It looks like a huge study with lots of findings. You tried to put lots of information including figures but too many figures with less explanation made me confused to continue my reading. In many parts of this manuscript, you started writing saying Fig X. Then you said Fig X, Y, Z. And suddenly started writing about the frame (a) and frame (b). But frames (a) and (b) from which figure - it is not clear. As a result, I could not see the main findings of this article. You should also write a para to summarize your main findings from this study. I will suggest using one para for 1 or 2 figures. Then another para for Figs 3 and 4. This will help readers to read your article with interest; otherwise will be confused like me. I am more than happy to review the revised copy suggesting a "minor revision". Thanks Reviewer

Round 2

Reviewer 3 Report

Dear Authors,

Of curse, this revised version is better than the previous one. However, I am still confused about some points in the result section when I match with figures to read the text.

First of all, some figures are too dense to see what is/are in the figure. For example, Fig 11. I am sorry, I am lost here and the description in the text is poor to explain the figure and its observations. Please fix them as there are some figures which are too dense. It is easy to make 2-4 subframes under one figure to make a better visualization.

Authors also need to write some points or para about the application of the outcomes, how this works could be helpful for the studies of dusty plasma systems or astrophysics or space science. Authors could read and may cite these articles if they get these useful:

a) Shock waves and double layers in electron degenerate dense plasma with viscous ion fluids

b) Roles of dust grains on electrostatic IA shocks in highly nonlinear dense plasma with degenerate electrons

The authors need to say about the limitations of this study and also suggest some possible ways to overcome that limitation for future studies if one could like to extend this study.

I would like to mark this time as "minor revision" and I am happy to revise the next revised version when it is ready.

Thanks

Reviewer

Round 3

Reviewer 3 Report

Dear Authors,

This version is really better than the previous version.

However, still, some points are missing, like (a) what were the limitations of this work? (b) how those could be overcome in the future if anyone likes to extend it? (c) does this study help model dusty plasma or studies of astrophysics/space science - (I) if Yes, then how? (ii) if No, then how could it be possible to develop a model?

It's a minor revision before a final decision is coming after the revised version.

thanks

Reviewer
